**Diurnal variations of NO$_2$ tropospheric vertical column density over the Seoul Metropolitan Area from the Geostationary Environment Monitoring Spectrometer (GEMS): seasonal differences and the influence of the *a priori* NO$_2$ profile**

Seunghwan Seo[1], Si-Wan Kim[1,2*], Kyoung-Min Kim[1], Andreas Richter[3], Kezia Lange[3], John P. Burrows[3], Junsung Park[4**], Hyunkee Hong[5], Hanlim Lee[4], Ukkyo Jeong[4], Jung-Hun Woo[6***], and Jhoon Kim[1*]

[1]Department of Atmospheric Sciences, Yonsei University, Seoul, Republic of Korea

[2]Irreversible Climate Change Research Center, Yonsei University, Seoul, Republic of Korea

[3]Institute of Environmental Physics, University of Bremen, Bremen, Germany

[4]Division of Earth Environmental System Science, Major of Spatial Information Engineering, Pukyong National University, Busan, Republic of Korea

[5]National Institute of Environmental Research, Incheon, Republic of Korea

[6]Department of Technology Fusion Engineering, College of Engineering, Konkuk University, Seoul, Republic of Korea

*To whom correspondence should be addressed:

Si-Wan Kim (e-mail: siwan.kim@yonsei.ac.kr) and Jhoon Kim (e-mail: jkim2@yonsei.ac.kr).

**now at: Center for Astrophysics | Harvard & Smithsonian, Cambridge, MA, USA

1   [***]now at: Graduate School of Environmental Studies, Seoul National University, Seoul,

2   Republic of Korea

3   Date: 10/13/2024

**Abstract**

The Geostationary Environment Monitoring Spectrometer (GEMS), launched in 2020,

provides both temporally and spatially continuous air quality data from geostationary

Earth orbit (GEO). This study first investigates the seasonal variations and diurnal

behavior of nitrogen dioxide ($NO_2$) tropospheric vertical column densities (TropVCDs)

over the Seoul Metropolitan Area (SMA) using GEMS data, retrieved by the IUP-UB

algorithm. We find that the magnitude of the $NO_2$ TropVCDs and its diurnal behavior

have significant seasonal dependences. In January, the highest $NO_2$ TropVCD values in

the range $27.5 - 28.9 \times 10^{15}$ molec. $cm^{-2}$ during the four seasons were observed at 15:00

local time (LT), and $NO_2$ TropVCD increases from the first retrieved values at 10:00

LT. On the other hand, we find the lowest values ($7.4 - 8.8 \times 10^{15}$ molec. $cm^{-2}$) are at

~14:00 LT in July. The VCD values in July increased up to 10:00 LT, then decreased

until 14:00 LT, bu then began to increase again. These different diurnal behaviors of

the TropVCDs in the different seasons reflect the differences in photochemical and

meteorological conditions as well as the emissions of NOx. Photochemical

transformations are typically more rapid in July and slower in January. The absolute

values and diurnal behavior of $NO_2$ TropVCDs are significantly influenced by the wind

speed, except in July. Moderate (wind speed $\geq$ 3m/s) or strong wind (wind speed > 5m/s)

reduced the magnitude of the diurnal behavior in January, implying that the $NO_2$ plumes

were transported downwind. Finally, we investigate the retrieved $NO_2$ TropVCDs with

that retrieved using different   *a priori* $NO_2$ data simulated by TM5 and WRF-Chem,

calculated using the most recent emission inventories. Although simulated VCDs from

WRF-Chem and TM5 show differences of up to a factor 2.75, retrieved $NO_2$ TropVCDs

using each *a priori* data have almost identical values and diurnal behaviors, except in

July. Notably, the diurnal behavior of the retrieved $NO_2$ TropVCDs are independent of

those from the two chemical transport models, indicating that observations of slant

column densities are the dominant factor in determining the diurnal behavior of $NO_2$

TropVCDs. Changes of the model horizontal resolution and volatile organic compounds

(VOC) emission inventory do not affect significantly the retrieved $NO_2$ TropVCDs in

1 this study. However, when the *a priori* $NO_2$ vertical profile was fixed as the values at

2 13:45 LT, the diurnal patterns of $NO_2$ TropVCDs showed significant changes with

3 differences of up to -18.3%.

## 1. Introduction

Nitrogen dioxide ($NO_2$) is one of the most important trace gases in the photochemical mechanisms, which determine the tropospheric distributions of ozone and secondary aerosol (Milford et al., 1989). Beginning with the launch of the passive remote sensing instrument GOME om ESA ERS-2 (Burrows et al., 1999) in 1995, then followed by SCIAMACHY on ESA Envisat in 2002 (Burrows et al., 1995 and Bovensmann et al., 1999), OMI on NASA AURA (Levelt et al., 2006), GOME-2 on ESA EUMETSAT Metop A, B and C (Callies et al., 2000, Munro et al., 2016), and TROPOMI on the ESA Sentinel 5 Precursor in 2018 (Veefkind et al., 2012), the amounts and distributions of stratospheric and tropospheric $NO_2$ vertical column densities (TropVCDs) have been retrieved at increasing spatial resolutions from these instruments, which all fly in sun-synchronous low earth orbit (LEO). By using the retrieved $NO_2$ TropVCDs from the LEO instruments, the tropospheric nitrogen oxide sources have been identified and their NOx emissions have been estimated, and the chemistry of the troposphere has been studied from the local to the global scale. While instruments on board LEO satellites provide spatially continuous data, observations are obtained only once or twice per day. It was recognized in the late 1990s that instruments similar to SCIAMACHY in geostationary orbit (GEO) would potentially deliver the diurnal variations of key trace gases (see the GeoTROPE concept in Burrows et al., 2004 and references therein). The measurements at the top of the atmosphere of the Geostationary Environment Monitoring Spectrometer (GEMS), launched in 2020, yield the first not only spatially but also temporally continuous air quality data over Asia from the geostationary orbit GEO (see Kim et al., 2020).

Mathematical inversion of the GEMS observations provides diurnal variations of the $NO_2$ TropVCD. These data products enable the seasonal changes not only in pollutant concentration but also in temporal characteristics, such as the times of the maxima and minima and the sources and sinks of $NO_2$, which vary by diurnally and

seasonally, to be studied for the first time from space.

As part of the differential optical absorption spectroscopy (DOAS) retrieval of $NO_2$ TropVCD data, air mass factors (AMF) are used to convert slant column density (SCD) to VCD. The assumptions used in the AMF calculation are explained in Richter and Burrows (2002) and Palmer et al (2001). In agreement with other studies, Lorente et al. (2017) reported that the AMF calculation is the largest source of error or uncertainty in $NO_2$ satellite retrievals. This is because of the assumption used to determine the ancillary or prior data used in the AMF calculation, such as surface albedo, terrain height, cloud parameters, and trace gas profiles. Consequently, the selection of optimal and appropriate *a priori* data is essential to accurately retrieve $NO_2$ TropVCDs from the observations of any nadir-sounding satellite spectrometer. This is in addition to the need to separate upper atmospheric $NO_2$ from that in the troposphere.

In this study we investigate two important issues using the GEMS $NO_2$ TropVCD data over the Seoul Metropolitan Area (SMA): (1) the influence of *a priori* profiles on the retrieved GEMS $NO_2$ TropVCDs and (2) the seasonal variation of the GEMS $NO_2$ TropVCD. In section 2 we describe the methods and data used.

Prior to our geophysical interpretation of the $NO_2$ TropVCD, in Section 3 we compared three GEMS datasets, retrieved with different *a priori* data from the WRF-Chem model. Thereby we investigated the influence of the inventories of the emissions of NOx, defined as the sum of nitrogen monoxide (NO) and nitrogen dioxide ($NO_2$) in an air mass, on the simulated and retrieved $NO_2$ TropVCD.

In Section 4, we utilized two chemical transport models (CTM), the Weather Research and Forecast model combined with Chemistry (WRF-Chem) and the global chemistry transport model TM5 (Tracer Model 5) to analyze both the seasonal variations and the influence of *a priori* $NO_2$ profiles. The seasonal changes in the magnitudes and the time of the maxima of the diurnal $NO_2$ TropVCD, which we define as the peak times, were investigated. The differences in the spatial distributions of $NO_2$

TropVCD between the WRF-Chem- and TM5-based GEMS datasets using different *a priori* data, were identified for each season and peak time. We also analyzed the influence of wind speed on the variations in the magnitude and diurnal behavior of the retrieved $NO_2$ TropVCDs.

## 2. Data and methods

### 2.1. GEMS products

GEMS is an ultraviolet-visible (UV-VIS) instrument, measuring contiguously the spectral range from 300 to 500 nm at a spectral resolution of ~ 0.6 nm (Kim et al., 2020). The nominal spatial resolution is 3.5 km × 7.7 km for gases including $NO_2$ data products. The overall field of regard (FOR) of GEMS covers 75° – 145°E longitude and 5°S – 45°N latitude. GEMS measures hourly during the daytime. The number of observations varies depending on the month, as a result of the length of the day and the measurement strategy. For South Korea, observations are least frequent in January, with six observations per day, and most frequent from April to September, with ten observations per day. We utilized GEMS $NO_2$ TropVCD data with the IUP-UB algorithm (GEMS IUP-UB products) in January, April, July, and October 2021 – detailed explanations of GEMS IUP-UB products are shown in Section 2.1.1.

2.1.1. GEMS IUP-UB products v1.0

The GEMS $NO_2$ vertical columns used in this study are from the scientific data product of the University of Bremen, version 1.0 (Lange et al., 2024, Richter et al., *in preparation*). $NO_2$ slant columns are retrieved in the large fitting window 405 – 485nm to reduce noise. In addition to the cross-sections of other absorbing species ($O_3$, $O_4$, $H_2O$ and liquid water) pseudo cross-sections for the Ring effect, for GEMS instrument polarization sensitivity and the effects of scene inhomogeneity are included. The

stratospheric correction is performed using the STRatospheric Estimation Algorithm from Mainz (STREAM) (Beirle et al., 2016). Conversion to vertical tropospheric columns is based on look-up tables of altitude dependent air mass factors calculated with the radiative transfer model SCIATRAN (Rozanov et al., 2014) using Lambertian equivalent reflectivity (LER) surface reflection values from the TROPOMI climatology (Tilstra et al., 2023). To apply the cloud correction, adjusted cloud fractions and pressure from the GEMS L2 cloud product were used. Further information about IUP-UB products is described in Richter et al. (*in preparation*). The $NO_2$ *a priori* data are different in the different model simulations, which we call runs, as explained below.

## 2.2. Experiment designs

To analyze the spatiotemporal characteristics of GEMS $NO_2$ VCDs and the impacts of different *a priori* data on the retrieved values, we undertook five experiments, called TM5, CTRL, CONST, FINE, and MIXED.

The TM5 experiment applies the standard GEMS IUP-UB products v1.0, which use the TM5 model, as their *a priori* data (Huijnen et al., 2010, Williams et al., 2017). The meteorological data for TM5 simulations are obtained from the European Centre for Medium-Range Weather Forecasts (ECMWF) operational forecast data. For the anthropogenic $NO_x$ emission inventory of TM5, the MACCity emission estimates are adopted (Granier et al., 2011), which have no diurnal variation of $NO_x$ emissions. The outputs from TM5 model have a horizontal resolution of $1° \times 1°$ and 34 vertical layers.

For the other four numerical experiments (CTRL, CONST, FINE, and MIXED), WRF-Chem version 4.4 was used to generate *a priori* data (Grell et al., 2005, Skamarock et al., 2021). The chemistry scheme follows the Regional Atmospheric Chemistry Mechanism (RACM) with Secondary Organic Aerosol-Volatility Basis Set

1 (SOA-VBS) option (chem_opt = 108) (Ahmadov et al., 2012). The horizontal

2 resolution of WRF-Chem simulation is 28 km × 28 km, except for the FINE run (12 km

3 × 12 km). All simulations have 59 customized vertical layers. To account for the

4 stratospheric vertical profiles, the Whole Atmosphere Community Climate Model

(WACCM) model outputs were combined with the WRF-Chem data

(ACOM/NCAR/UCAR, 2020, last access: 05 Dec 2022). The combined data comprises

a total of 113 vertical layers. Detailed model configuration is described in Kim et al.

(2024). For the anthropogenic emission inventories, the Air Quality in Northeast Asia

(AQNEA) emission inventory version 2 was adopted. Since the reference year of

AQNEA version 2 is 2019, the anthropogenic $NO_x$ emissions decreased by 20% to

account for the decreasing trends of $NO_x$ emissions from 2019 to 2021. We applied the

normalized diurnal variabilities of $NO_x$ emissions obtained from the Los Angeles Basin

in Kim et al. (2016), but shifting the values one hour earlier (**Figure 1**). For the CONST

run, only the *a priori* profiles at 13:45 LT were used to retrieve the $NO_2$ TropVCD. To

investigate the impact on the volatile organic compounds (VOC) emissions of the

anthropogenic VOC emissions we used the KORUS emission inventory version 5 (Jang

et al., 2020, Woo et al., 2012) in the MIXED run. We retrieved four months (January,

April, July, and October 2021) for the TM5 and CTRL runs, and one month (July 2021)

for the other runs. The experimental designs are summarized in **Table 1**.

**3. Impacts of different *a priori* data on the retrieved $NO_2$ TropVCDs**

We compared retrieved $NO_2$ TropVCDs from the five different simulations, or runs, to

study the impacts of *a priori* data used in AMF calculations on the retrieved $NO_2$

TropVCD.

**3.1. Comparison between the CTRL and TM5 runs**

Retrieved $NO_2$ TropVCDs from the CTRL and TM5 runs exhibit similar diurnal

patterns, which are independent of the diurnal patterns of their respective *a priori* data (**Figure 2**). This suggests that the observed slant column density (SCD) plays a more decisive role in the diurnal pattern of TropVCD than the influence of *a priori* used to determine the AMF. Nevertheless, differences in $NO_2$ TropVCDs between the two runs were observed, and are particularly noticeable differences in July.

**Figure 3** displays spatial distributions of AMF differences between the CTRL and TM5 runs in January, April, July, and October 2021. In urban areas, the AMF in the CTRL run was generally lower (blue) than in the TM5 run, but higher values (red) were observed in the northern and eastern regions of Seoul. As a result, the average values across the SMA domain were similar between CTRL and TM5 – the diurnal patterns of averaged air mass factor over the SMA are shown in **Figure 4**. In July, however, lower values in the CTRL run were observed throughout Seoul and its surrounding areas, leading to lower average AMF values for the SMA region during most of the day. As a result, the TropVCD values in July were higher in the CTRL run (Figure 2c).

In **Figure 5**, we compare $NO_2$ vertical profiles at 08, 10, 12, 14, and 16 LT from the CTRL and TM5 runs. $NO_2$ values in the lower atmosphere in the CTRL run are much higher than those in the TM5 run in July, which lead to lower AMF and thus higher $NO_2$ TropVCD.

**3.2. Comparisons between the CTRL and CONST, FINE, and MIXED runs**

In **Figure 6**, the diurnal patterns of retrieved and *a priori* $NO_2$ TropVCDs in July 2021 over the SMA region from the CTRL run and the CONST, FINE, and MIXED runs, are shown. Despite some changes in model resolution and VOC emissions, the FINE and MIXED runs did not show significant differences compared to the CTRL run. In particular, the MIXED run resulted in almost no difference in the *a priori* TropVCD , resulting in nearly identical retrieved $NO_2$ TropVCD.

On the other hand, the CONST run, which used only the *a priori* vertical profile
from 13:45 LT in the retrieval process, exhibited clear differences to the CTRL run.
Specifically it had lower values than the CTRL run before ~14:00 LT, but higher values
after. These differences are explained by comparisons of vertical profiles from each run,
which are displayed in **Figure 7**. The vertical profile shapes of the CTRL, FINE, and
MIXED runs are identical, indicating that AMF of each runs have similar values. On
the other hand, clear differences of vertical profile shape are apparent between the
CTRL and CONST runs. Before 14:00 LT, the CTRL run showed lower sensitivity in
the upper layers compared to the CONST run. This indicates a smaller AMF and thus
higher VCD values. In contrast, after 14:00 LT, the CTRL run exhibited higher
sensitivity to $NO_2$ in the upper layers of the troposphere, leading to a larger AMF and
consequently lower VCD values compared to the CONST run. These differences in the
vertical profile arise from effects such as the development of the mixing layer and
variations in emissions throughout the day. This implies that providing optimal time-
dependent *a priori* data for the AMF calculation will improve the accuracy of the
retrieved $NO_2$ TropVCD.

## 4. Spatiotemporal characteristics of GEMS $NO_2$ TropVCD

We report on our investigation of the spatiotemporal characteristics of GEMS $NO_2$
TropVCD. We use the retrieved $NO_2$ TropVCD and those simulated by the TM5 and
CTRL runs to assess two geophysically important influences on the $NO_2$ TropVCD the
SMA region (126.5 – 127.3°E, 37.2 – 37.8°N) in 2021: (1) the identification,
quantification and origin of the seasonal changes; and (2) advection and convection of
air masses.

### 4.1. Seasonal variations

**Figure 2** displays diurnal patterns of retrieved and *a priori* $NO_2$ TropVCDs during

weekdays in January, April, July, and October 2021 over the SMA region from the TM5 and CTRL runs. The scenes with wind speed faster than 3m/s are excluded to remove the transport impacts. The effects of transport on $NO_2$ columns are analyzed in **Section 4.2**.

In January, $NO_2$ TropVCDs continuously increase from 10:00 local time (LT) to 15:00 LT. During the winter, $NO_2$ in the urban region accumulates particularly in the boundary layer. Qualitatively, this is explained as follows.

As tropospheric solar UV radiation is low in winter and the atmosphere is cold, photolysis frequencies are small. Similarly, the rate coefficients of many reactions are smaller at the lower winter temperatures compared to those of the other seasons. In winter, the relatively slow loss of NOx occurs through the three body reaction of hydroxyl (OH) with $NO_2$ to form nitric acid ($HNO_3$):

$$OH + NO_2 + M \rightarrow HNO_3 + M. \qquad (1)$$

The smaller photolysis frequencies of reactions following photoexcitation in the reactions:

$$NO_2 + hv(\lambda < 405 \text{ nm}) \rightarrow NO + O \qquad (2)$$

$$O_3 + hv(\lambda < 405 \text{ nm}) \rightarrow O(^1D) + O_2 \qquad (3)$$

lead to slower production of i) the first excited state of oxygen ($O(^1D)$) from the photolysis of ozone ($O_3$), ii) the hydroxyl radical (OH), and iii) the production of organic peroxyl radicals ($RO_2$), and hydroperoxyl ($HO_2$) through the oxidation of methane ($CH_4$) and VOC. Some of the following reactions are involved:

$$O + O_2 + M \rightarrow O_3 + M \qquad (4)$$

$$O(^1D) + N_2 \rightarrow O + N_2 \qquad (5)$$

$O(^1D) + O_2 \rightarrow O + O_2$       (6)

$O(^1D) + H_2O \rightarrow OH + OH$       (7)

$OH + CH_4 \rightarrow CH_3 + H_2O$       (8)

$CH_3 + O_2 + M \rightarrow CH_3O_2 + M$       (9)

$CH_3O_2 + NO \rightarrow CH_3O + NO_2$       (10)

$CH_3O + O_2 \rightarrow HO_2 + HCHO$       (11)

$OH + VOC + O_2 \rightarrow nHO_2 + aldehydes$       (12)

$OH + CO + O_2 \rightarrow HO_2 + CO_2$       (13)

$HO_2 + NO \rightarrow OH + NO_2$       (14)

Overall at low solar insolation, the low levels of actinic radiation result in

smaller amounts of OH and $HO_2$. The oxidation process is slow and $HO_2$ and OH

chemistry are coupled with NOx chemistry and controlled by rate of oxidation of VOC

and $CH_4$ and the rate of $HO_2$ to OH through the rate of reaction (14) and the rate of loss

of HOx and NOx for example through reaction (1).

In January the maximum values of retrieved TropVCDs are $27.5 \times 10^{15}$ molec.

$cm^{-2}$ (TM5) and $28.9 \times 10^{15}$ molec. $cm^{-2}$ (CTRL) at 15:00 LT, whereas the *a priori* $NO_2$

TropVCDs have maxima of $11.2 \times 10^{15}$ molec. $cm^{-2}$ (TM5) and $21.9 \times 10^{15}$ molec. $cm^{-}$

$^2$ (CTRL) at the same time. These higher values of retrieved $NO_2$ TropVCDs relative to

the model $NO_2$ TropVCDs are explained by the following inadequate knowledge of the

bottom-up diurnal $NO_x$ emissions in January and/or the dilution during the transport of

plumes, which is dependent on the model horizontal resolution.

For other months, the maxima of $NO_2$ TropVCDs occur at earlier times of the day in April at 12:00 LT, in July at 10:00 LT and in October at 11:00 LT. There is also a second maximum at 15:00 LT in October. The behavior of $NO_2$ TropVCD in April, July and October, when compared to that in January, is explained by the following effects: i) faster tropospheric photolysis frequencies, as a result of higher levels of tropospheric solar insolation and actinic radiation accelerating the photochemical oxidation of $CH_4$ and VOC in April, July and October compared to January; ii) generally faster reaction rate coefficients of the free radical reactions at the higher temperatures, the rate coefficient of reaction (4) being an exception; iii) the different diurnal emissions of NOx compared to those in January. **Figure 8** shows the diurnal variations of OH concentrations averaged across boundary layer height in each month, calculated by the CTRL model run. The OH concentration in January is about an order of magnitude smaller than that in July.

In April, $NO_2$ TropVCDs increased until 12:00 LT. It then maintains similar levels until 17:00 LT. The maximum $NO_2$ TropVCD occurred at 12:00 LT for the CTRL run ($21.4 \times 10^{15}$ molec. $cm^{-2}$). The maximum $NO_2$ TropVCD for the TM5 run appeared at 17:00 LT, being $21.9 \times 10^{15}$ molec. $cm^{-2}$. However, the retrieved $NO_2$ TropVCDs from the TM5 and CTRL runs have almost identical behavior up to 15:00 LT. There is a difference of $1.6 \times 10^{15}$ molec. $cm^{-2}$ (8.1%) between the two runs at 17:00 LT, when *a priori* $NO_2$ TropVCD value sharply increased from the CTRL run.

In July, both the TM5 ($12.2 \times 10^{15}$ molec. $cm^{-2}$) and CTRL ($13.9 \times 10^{15}$ molec. $cm^{-2}$) runs show maxima at 10:00 LT, i.e. the earliest for the four months investigated. After the peak, $NO_2$ TropVCDs decrease, most likely due to more rapid photochemical loss processes e.g. reaction (1) until 14:00 LT, and then increase. In other seasons, the minimum values were observed in the morning. However, in July, the minimum occurred at 14:00 LT. This unique pattern of behavior is explained by the more rapid

photochemical production and removal reactions in summer. We infer that the chemical

removal becomes relatively more rapid than the emission and production of $NO_2$ (see

Figure 8). The two types of run show similar diurnal behavior, but the retrieved $NO_2$

TropVCD of the CTRL runs between 10:00 and 14:00 LT rise to $2.1 \times 10^{15}$ molec. cm$^-$

$^2$ i.e. higher than those of the TM5 runs. The diurnal change of *a priori* $NO_2$ TropVCDs

from the CTRL runs shows a similar behavior to that of the retrieved $NO_2$ TropVCDs,

despite the magnitude of *a priori* $NO_2$ TropVCD being $3.9 - 8.2 \times 10^{15}$ molec. cm$^{-2}$

higher than those retrieved. On the other hand, the *a priori* $NO_2$ TropVCD from the

TM5 runs decreases between 08:00 and 14:00 LT, reflecting diurnally varying

photochemistry with similar levels of $NO_x$ emissions throughout the day.

In October, there are broad maxima of $NO_2$ TropVCD between 12:00 LT and

15:00 LT. Overall diurnal behavior comprises increase up to 12:00 LT, followed by

broad maxima, after which the $NO_2$ TropVCD are similar to those in April.

The highest retrieved values are in the range $25.1 \times 10^{15}$ to $25.5 \times 10^{15}$ molec.

cm$^{-2}$ for both the TM5 and CTRL runs. As expected, the $NO_2$ TropVCD are the highest

in January and lowest in July.

**Figures 9** and **10** show the spatial distributions of retrieved $NO_2$ TropVCDs in

January, April, July, and October 2021 from the TM5 and CTRL runs, respectively. In

January, a plume over the SMA region developed as a function of time. Consequently,

the suburban areas, which surround the SMA region, experience relatively high $NO_2$

TropVCD ($> 10 \times 10^{15}$ molec. cm$^{-2}$) compared to that retrieved in the other months. In

April and October, the plumes over the SMA are saturated prior to 12:00 LT and then

decrease. In contrast, the $NO_2$ TropVCD of the surrounding regions are relatively

constant or even increase. In July, the overall low values cover the SMA and nearby

regions for whole days. The maximum values appear at 10 LT, and then decreased until

14 LT. However, the $NO_2$ VCD rebounded at 16 LT. **Figure 11** displays the differences

between the TM5 and CTRL runs – red color indicates the CTRL run has higher values

than the TM5 run; blue means opposite. The CTRL run shows higher VCD values than

the TM5 run for all times in July. The largest differences over the SMA region are found

at 12 LT in July with differences of $2.1 \times 10^{15}$ molec. cm$^{-2}$. In other months, the CTRL

run generally have higher values of VCD than the TM5 run over the Seoul and urban

regions, while there are lower values of VCD from the CTRL run over rural regions.

## 4.2. Impacts of horizontal transport

**Figure 12** shows the diurnal behavior of the retrieved NO$_2$ TropVCDs from the CTRL

run for different wind conditions. The black lines indicate calm runs (wind speed lower

than 3m/s), the green line is a strong-wind run (wind speed faster than 5m/s), and the

blue lines are the average values with no wind filters. In January (Figure 12a), the

diurnal behavior of the NO$_2$ TropVCDs change significantly with the wind conditions.

In the calm run (black solid), NO$_2$ TropVCD steadily increases due to a combination of

the emissions increasing and the slow chemical loss in this month. In windy runs,

however, diurnal changes in the retrieved NO$_2$ TropVCD are negligible.

Although the chemical loss is slow during wintertime, the accumulation of NO$_2$

was mitigated as strong winds transported large concentrations of NO$_2$ to downwind

regions. The differences between calm and other runs were most significant at 15:00

LT, further indicating that continuous outflow due to transport suppressed the

accumulation. As the wind speed increased, there was a noticeable reduction in NO$_2$

TropVCD values, which indicates a clear inverse relationship between wind speed and

VCDs, as shown in Edwards et al. (2024). The values of calm, average (blue solid), and

strong wind (green solid) are 19.0 – 28.9, 17.2 – 19.8, and 12.1 – 13.4 $\times 10^{15}$ molec.

cm$^{-2}$, respectively.

In April (Figure 12b) and October (Figure 12d), the averaged values with no

wind filters (blue solid) have different diurnal behavior, but the maximum NO$_2$

TropVCD appear almost simultaneously with that of the calm low wind speed run. In July (Figure 7c), however, the diurnal behavior from the calm run and no wind filters are nearly identical, implying that the wind speeds are overall slow in July.

In summary, the transport effect is maximized in wintertime, changing not only the absolute values but also diurnal behavior of $NO_2$ Trop VCDs. Consequently, transport must be taken into account when analyzing $NO_2$ TropVCDs and when estimating top-down $NO_x$ emissions. The role of transport needs to be taken into account even for cases, where the wind speed is relatively slow during summertime (Yang et al., 2024).

**5. Conclusions**

In this study, we analyzed the seasonal variations and diurnal behavior of the retrieved GEMS IUP-UB $NO_2$ TropVCD, using the monthly mean data in January, April, July and October. The effects of wind speed, and the impact of *a priori* $NO_2$ profiles on the retrieval. Both in the CTRL and TM5 runs, the GEMS $NO_2$ product showed significant changes in quantity, diurnal pattern, and peak time as the seasons changed. In winter, the values were the highest, with a gradual increase over time, whereas in summer, the values were the lowest, reaching a minimum in the afternoon. This is consistent with previous studies, which have shown that atmospheric chemical reactions are more active in summer. Furthermore, we confirmed that wind-driven transport significantly influences the diurnal patterns, clearly demonstrating that advection and possibly convection need to be taken into account when top-down NOx emissions are estimated from an urban agglomeration such as SMA.

On the other hand, when using different *a priori* data to calculate VCD values, more complex results emerged. A comparison between the CTRL and TM5 runs revealed that, despite different spatial resolution and emission characteristics, the

retrieved $NO_2$ TropVCDs exhibited similar diurnal patterns, with significant differences only in July. Additionally, we found that the retrieved $NO_2$ TropVCDs had diurnal behaviors independent of the *a priori* data in both runs. We infer that the observed SCD has a stronger influence on the retrieved diurnal patterns than *a priori* profiles. Adjusting the horizontal resolution of the model (FINE run) or changing the VOC emissions data (MIXED run) also resulted in no significant differences. However, in the CONST run, where only the vertical profile at 14:00 LT was used in the retrieval process throughout the day, there were significant differences in both the $NO_2$ Trop VCD values and diurnal patterns. This reaffirms that the vertical shape factor of *a priori* data plays a critical role in $NO_2$ TropVCD retrievals.

Additionally, given that vertical as well as horizontal model resolution can influence retrievals (Liu et al., 2020), future studies should analyze the results when the vertical resolution is adjusted. Furthermore, as highlighted by previous studies, such as Lorente et al. (2017) and Hong et al. (2017), which emphasized the importance of cloud parameters, aerosol characteristics, and surface albedo, uncertainties arising from factors in addition to the *a priori* $NO_2$ profile should further be investigated in the retrieval of $NO_2$ TropVCD for both diurnal GEO observation and those from LEO.

**Data availability**

GEMS measurement data retrieved by the IUP algorithm are available on request from Andreas Richter (richter@iup.physik.uni-bremen.de). WRF-Chem v4.4 is available in GitHub (wrf-model, 2022).

**Author contributions**

SWK initiated this study and secured funding. SS and SWK analyzed the satellite and model data. SS, KMK, and SWK conducted the model simulations. AR, KL, and JPB

provided GEMS IUP products and analyzed the data. JK, JP, HH, HL, UJ retrieved and analyzed the GEMS observations and discussed the results. JHW provided AQNEA version 2 emission inventory. SS and SWK wrote the paper, with contributions from all co-authors.

**Competing interests**

At least one of the authors is a member of the editorial board of Atmospheric Measurement Techniques.

**Acknowledgement**

This work was supported by the National Research Foundation of Korea (NRF) grant funded by the Korea government (MSIT) (No. 2020R1A2C2014131). All the computing resources are provided by National Center for Meteorological Supercomputer. Th contributions from the University of Bremen were supported by the State and University of Bremen and the DLR.

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

**List of Figures**

**Figure 1.** Diurnal variabilities of normalized NOx emissions for CTRL (black) and TM5 (gray) runs over the SMA region.

**Figure 2.** Diurnal behavior of retrieved (solid) and *a priori* (dashed) $NO_2$ TropVCDs during weekdays in (a) January, (b) April, (c) July, and (d) October 2021 over the SMA region. Gray lines identify the TM5 run, while black lines represent the CTRL run. The pixels with wind speed faster than 3m/s are excluded.

**Figure 3.** Spatial distributions of air mass factor (AMF) differences (CTRL – TM5) in January, April, July, and October 2021. The pixels with wind speed faster than 3m/s are excluded.

**Figure 4.** Diurnal patterns of the air mass factor during weekdays in (a) January, (b) April, (c) July, and (d) October 2021 over the SMA region. Gray lines indicate the TM5 run, while black lines mean the CTRL run. The pixels with wind speed faster than 3m/s are excluded.

**Figure 5.** Vertical profiles of *a priori* $NO_2$ mixing ratios at 08, 10, 12, 14, and 16 LT from the TM5 (gray) and CTRL (black) runs in January, April, July, and October 2021 over the SMA region.

**Figure 6.** Diurnal patterns of retrieved (solid) and *a priori* (dashed) $NO_2$ TropVCDs in July 2021 over SMA region from the CTRL run (black) and (a) CONST run (red), (b) FINE run (pink), and (c) MIXED run (yellow). The pixels with wind speed faster than 3m/s are excluded. Note that diurnal changes of *a priori* $NO_2$ TropVCDs in the CONST run occur during calculating domain-averaged values – the location and number of pixels excluded during the collocation with satellite data vary over time during the day.

**Figure 7.** Vertical profiles of *a priori* $NO_2$ mixing ratios at 08, 10, 12, 14, and 16 LT from the CTRL (black), CONST (red), FINE (pink), and MIXED run (yellow) in January, April, July, and October 2021 over the SMA region.

**Figure 8.** Diurnal patterns of boundary layer mean OH concentrations over the SMA region in January (black), April (yellow), July (red), and October (blue) 2021 from the CTRL run.

**Figure 9.** Spatial distributions of retrieved $NO_2$ TropVCDs in January, April, July, and October 2021 taking the a priori data for the AMF form the TM5 run. The scenes with wind speed faster than 3m/s are excluded to minimize the impact of rapid transport.

**Figure 10.** Same as Figure 9, except that *a priori* values for the AMF calculation are taken from the CTRL run.

**Figure 11.** Similar to Figure 9, but for the differences of $NO_2$ TropVCD between CTRL and TM5 run (CTRL – TM5).

**Figure 12.** Diurnal patterns of retrieved $NO_2$ TropVCDs from the CTRL run in (a) January, (b) April, (c) July, and (d) October 2021 over the SMA region. Black lines indicate the NO2 TropVCD values with wind-filtered data; only the scenes with wind speed lower than 3m/s are utilized. Blue lines are the averaged values without any wind filters. The green line is for case of strong-wind run with the $NO_2$ TropVCD being selected and averaged for wind speeds faster than 5m/s in January.

1    **Table 1.** Description of the experimental designs. MACCity provides hourly-constant

2    emissions, while the others provide hourly-varying emissions.

| Run name | Model | Horizontal resolution | Emission inventory |
|---|---|---|---|
| TM5 | TM5 | 1° × 1° | MACCity |
| CTRL | | 28 × 28 km$^2$ | 2021AQNEA |
| CONST[a] | WRF-Chem v4.4 | 28 × 28 km$^2$ | 2021AQNEA |
| FINE | | 12 × 12 km$^2$ | 2021AQNEA |
| MIXED | | 28 × 28 km$^2$ | (VOC) KORUSv5 (others) 2021AQNEA |

4    [a] CONST run uses hourly-varying emission inventory, but only data of 13:45 LT were
5    utilized to compute AMF.

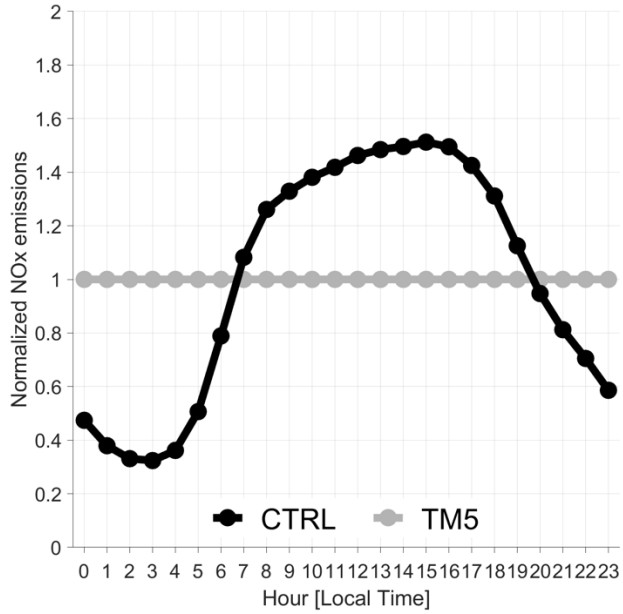

2    **Figure 1.** Diurnal variabilities of normalized NOx emissions for CTRL (black) and

3    TM5 (gray) runs over the SMA region.

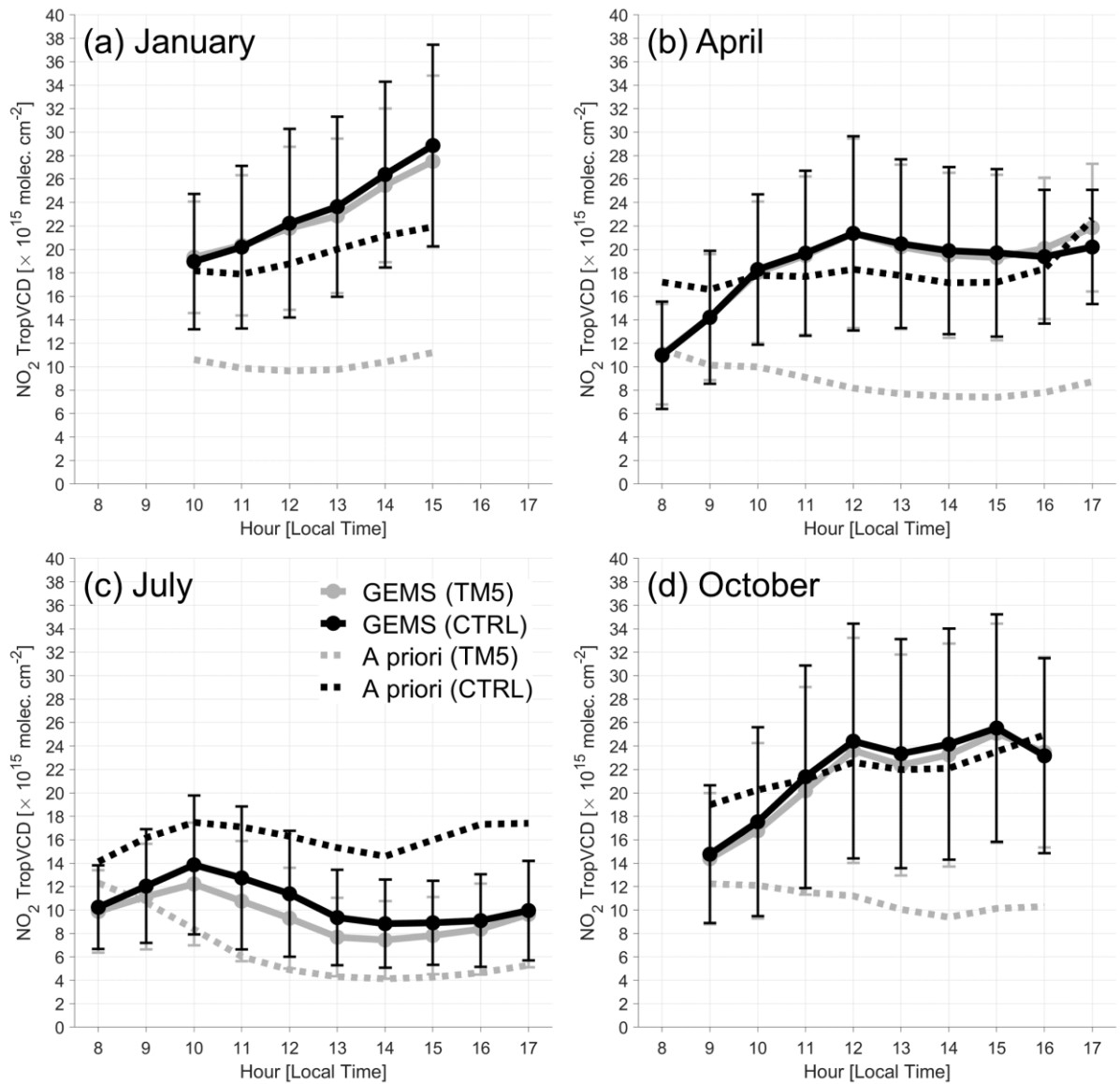

**Figure 2.** Diurnal behavior of retrieved (solid) and *a priori* (dashed) NO₂ TropVCDs during weekdays in (a) January, (b) April, (c) July, and (d) October 2021 over the SMA region. Gray lines identify the TM5 run, while black lines represent the CTRL run. The pixels with wind speed faster than 3m/s are excluded.

- 31 -

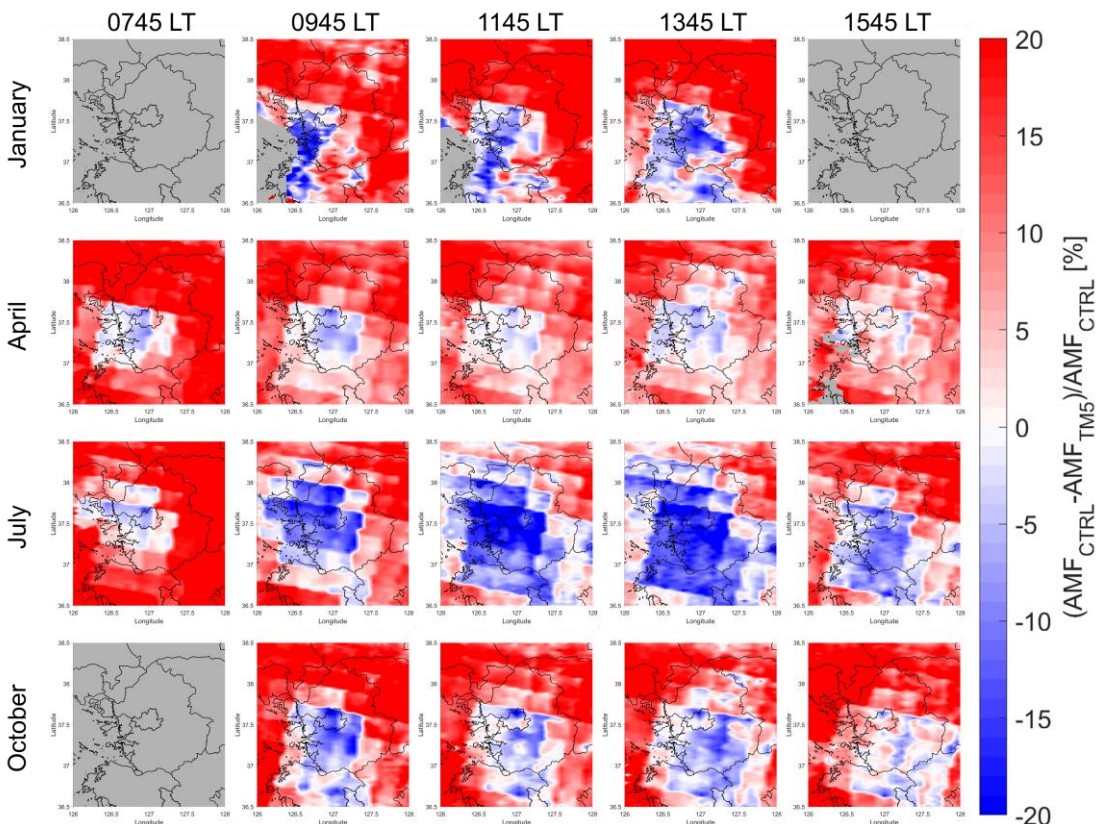

Figure 3. Spatial distributions of air mass factor (AMF) differences (CTRL – TM5) in January, April, July, and October 2021. The pixels with wind speed faster than 3m/s are excluded.

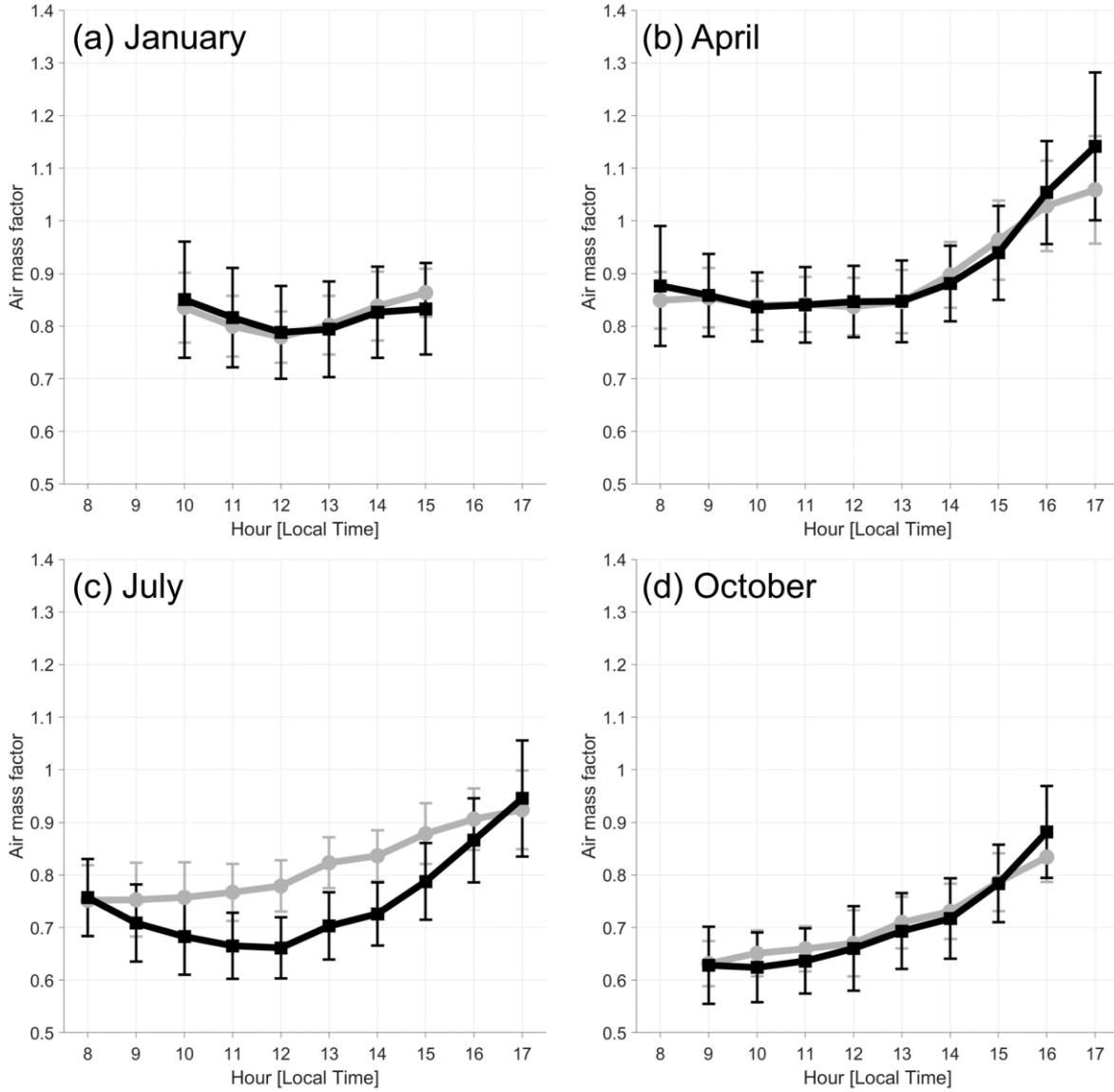

**Figure 4.** Diurnal patterns of the air mass factor during weekdays in (a) January, (b) April, (c) July, and (d) October 2021 over the SMA region. Gray lines indicate the TM5 run, while black lines mean the CTRL run. The pixels with wind speed faster than 3m/s are excluded.

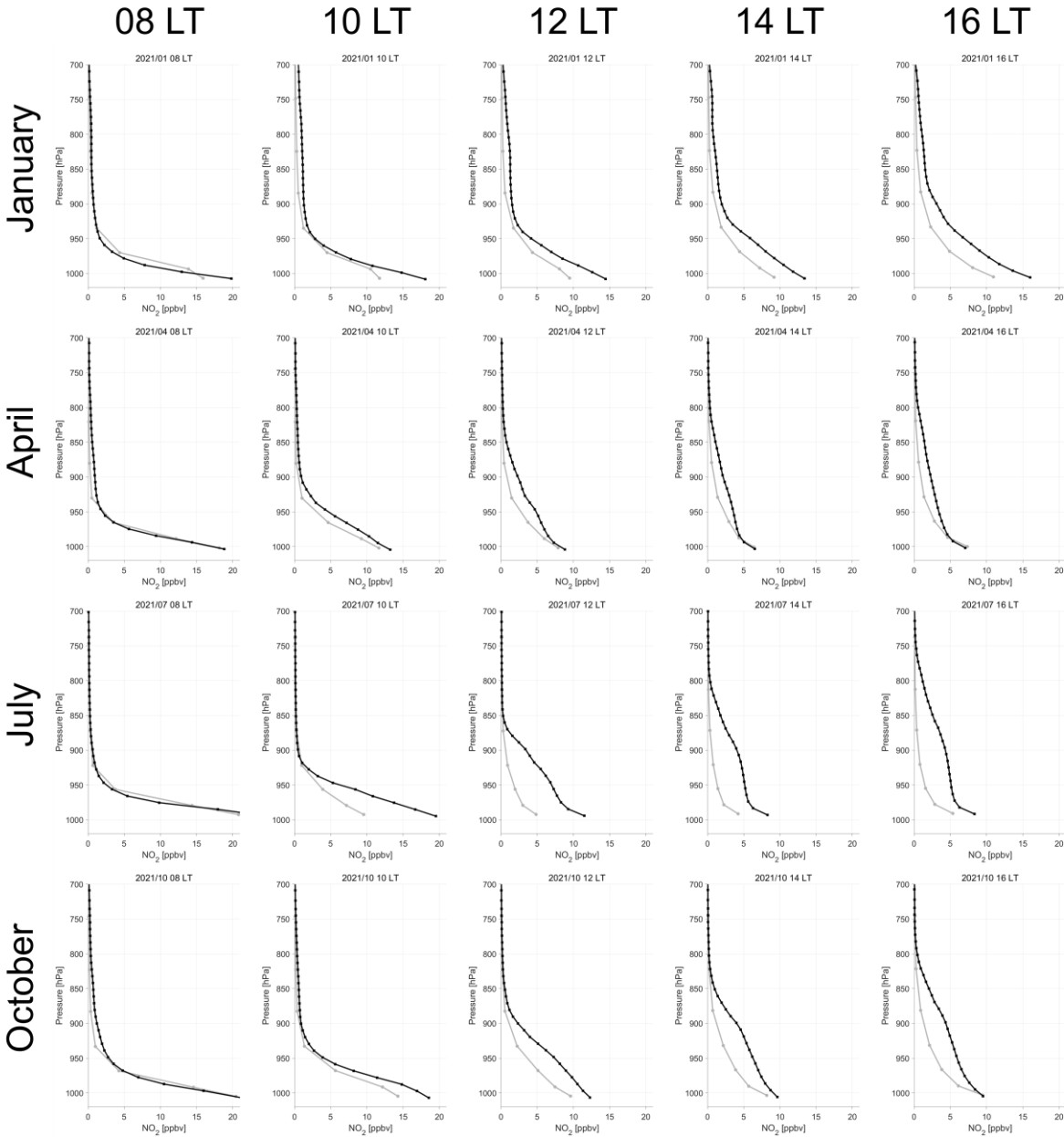

**Figure 5.** Vertical profiles of *a priori* NO₂ mixing ratios at 08, 10, 12, 14, and 16 LT

from the TM5 (gray) and CTRL (black) runs in January, April, July, and October 2021

over the SMA region.

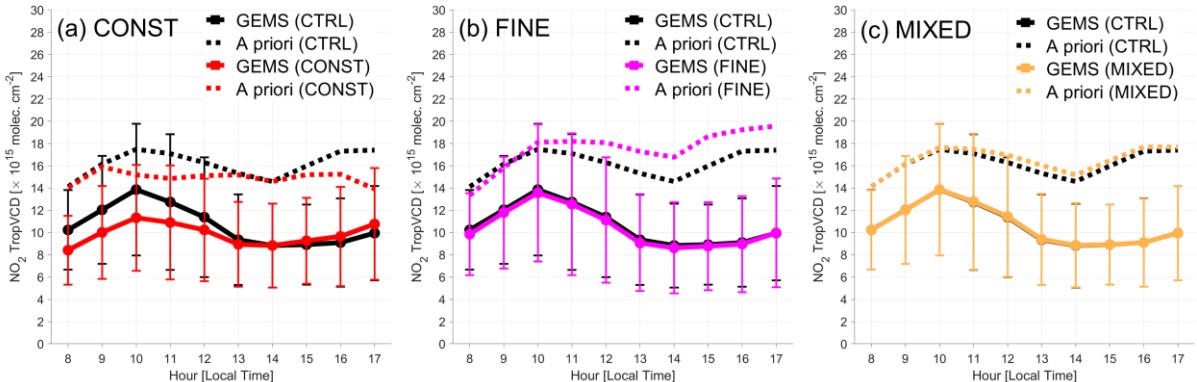

**Figure 6.** Diurnal patterns of retrieved (solid) and *a priori* (dashed) NO$_2$ TropVCDs in

July 2021 over SMA region from the CTRL run (black) and (a) CONST run (red), (b)

FINE run (pink), and (c) MIXED run (yellow). The pixels with wind speed faster than

3m/s are excluded. Note that diurnal changes of *a priori* NO$_2$ TropVCDs in the CONST

run occur during calculating domain-averaged values – the location and number of

pixels excluded during the collocation with satellite data vary over time during the day.

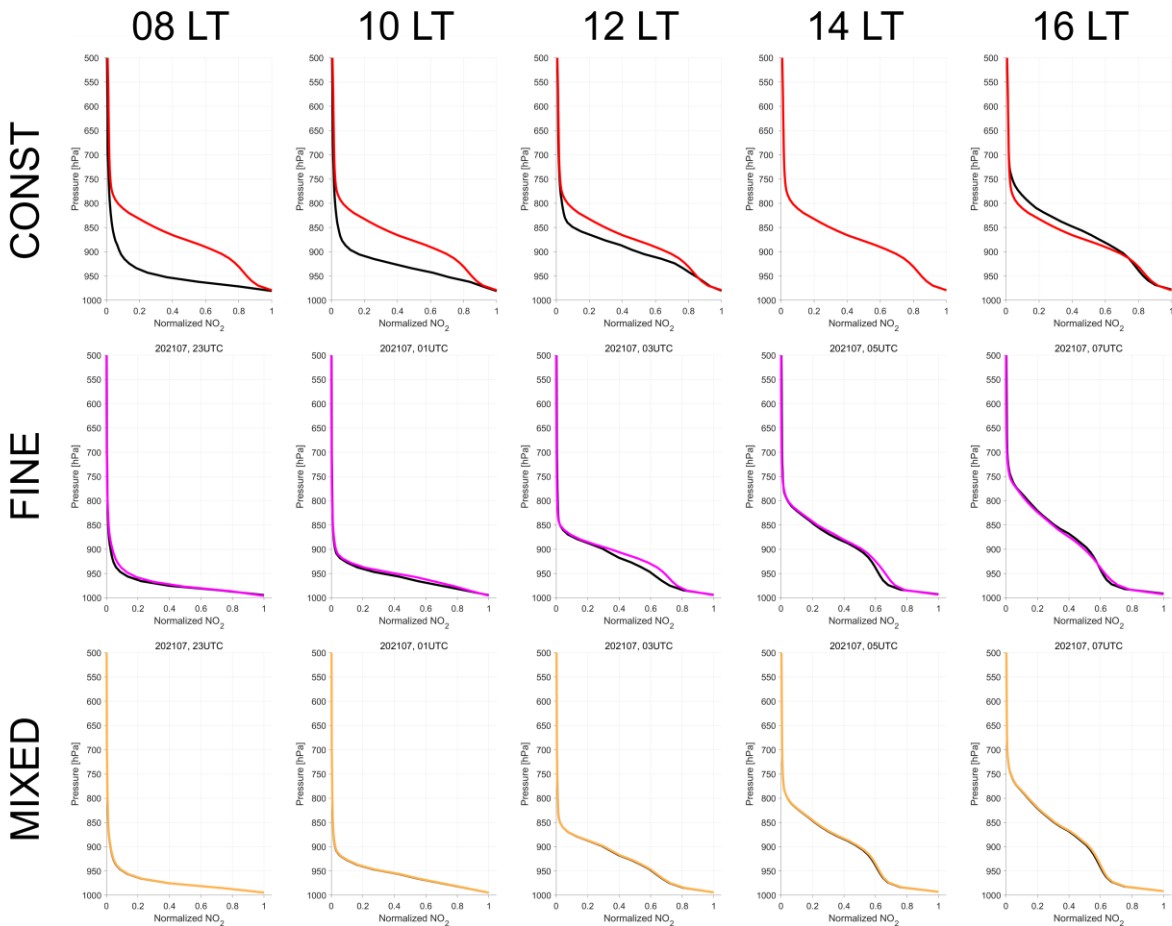

Figure 7. Vertical profiles of *a priori* NO$_2$ mixing ratios at 08, 10, 12, 14, and 16 LT from the CTRL (black), CONST (red), FINE (pink), and MIXED run (yellow) in January, April, July, and October 2021 over the SMA region.

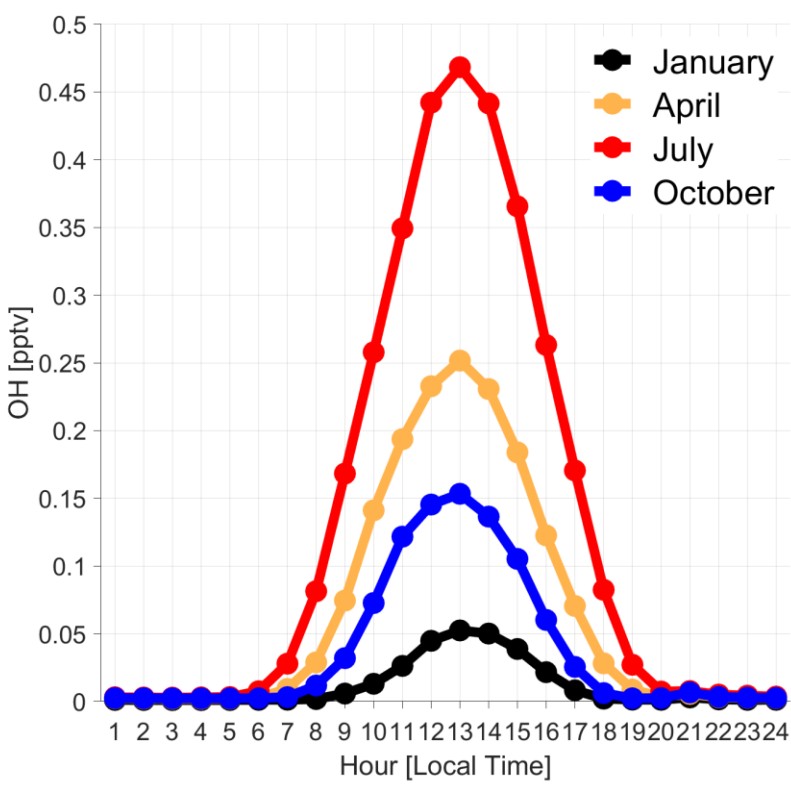

2 **Figure 8.** Diurnal patterns of boundary layer mean OH concentrations over the SMA

3 region in January (black), April (yellow), July (red), and October (blue) 2021 from the

4 CTRL run.

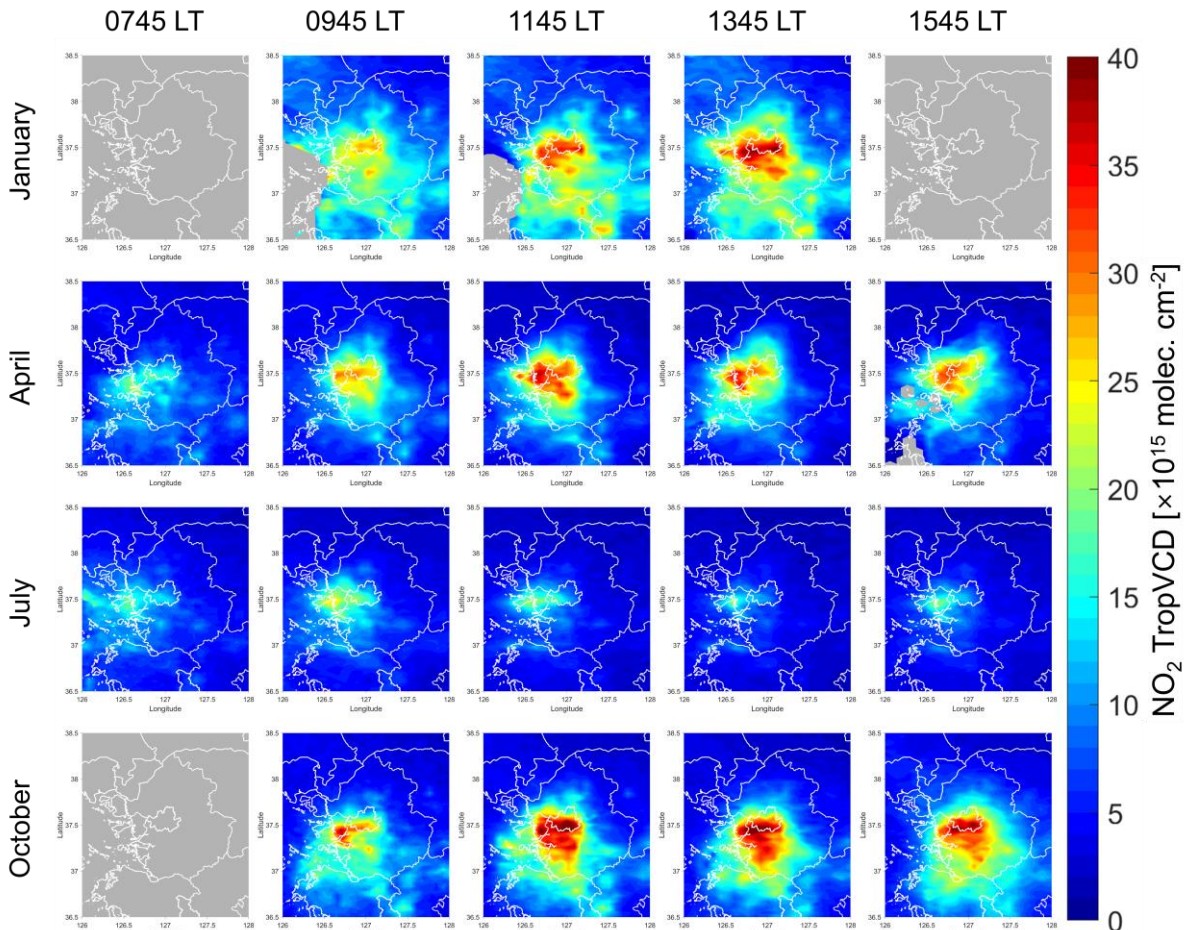

**Figure 9.** Spatial distributions of retrieved NO₂ TropVCDs in January, April, July, and October 2021 taking the a priori data for the AMF form the TM5 run. The scenes with wind speed faster than 3m/s are excluded to minimize the impact of rapid transport.

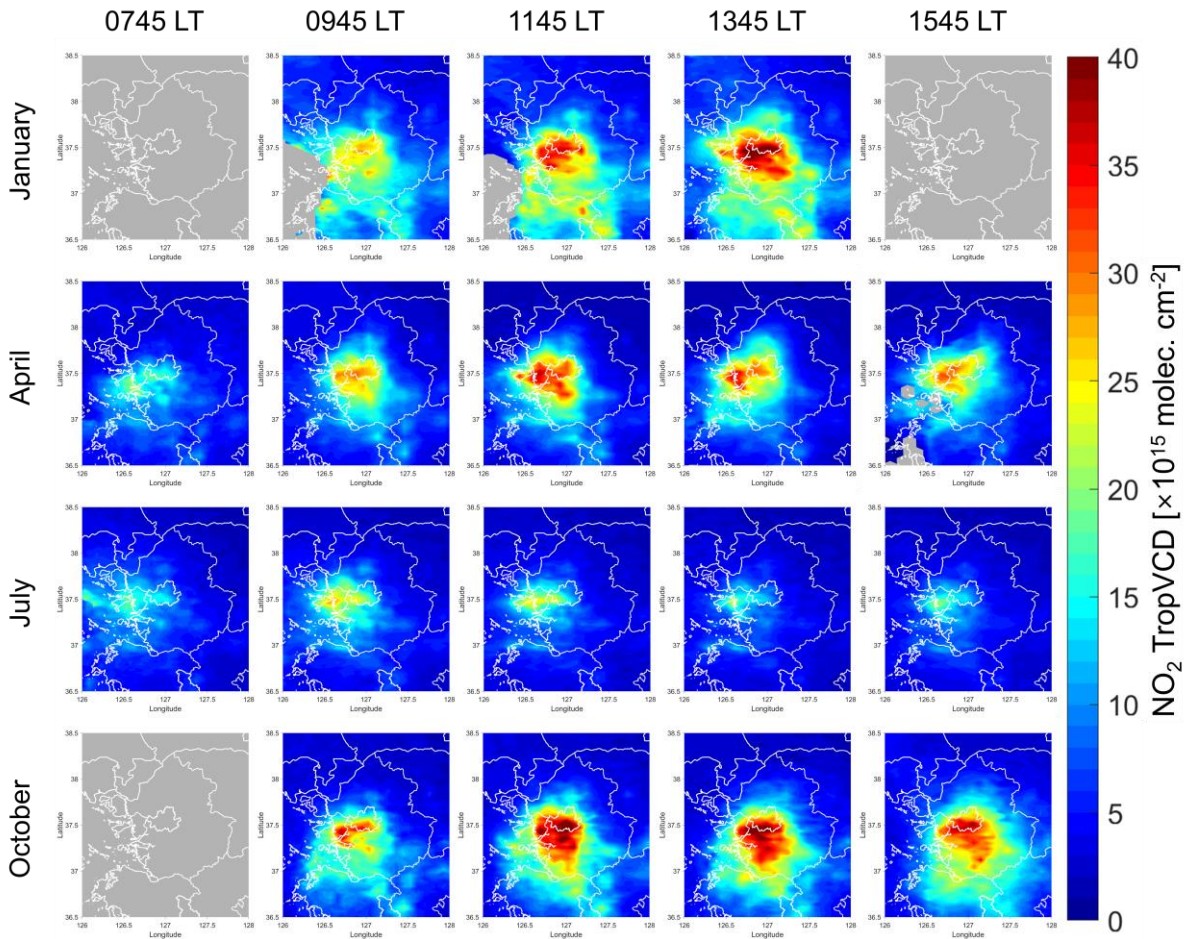

Figures with column headers 0745 LT, 0945 LT, 1145 LT, 1345 LT, 1545 LT and rows January, April, July, October. Color bar: NO$_2$ TropVCD [×10$^{15}$ molec. cm$^{-2}$].

**Figure 10.** Same as Figure 9, except that *a priori* values for the AMF calculation are taken from the CTRL run.

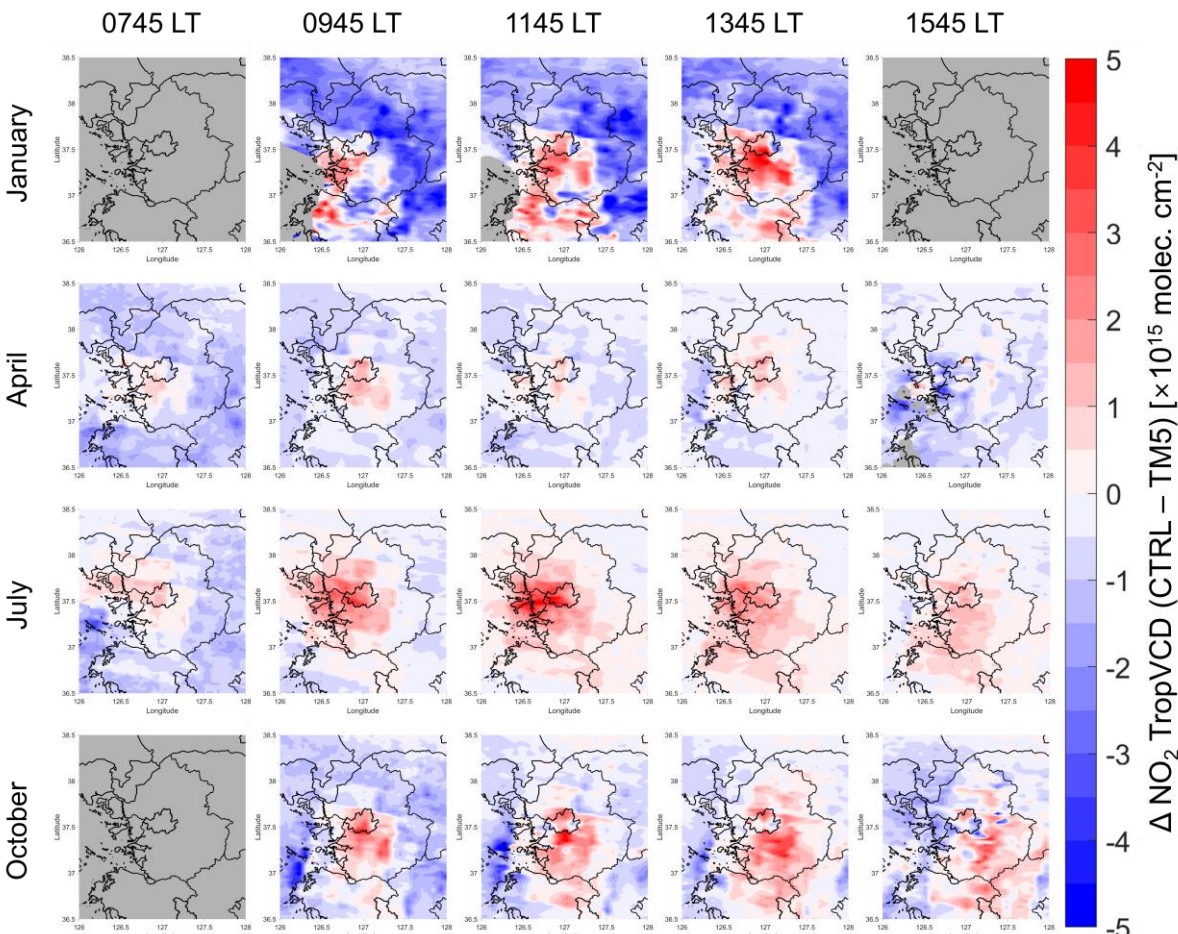

Figure 11. Similar to Figure 9, but for the differences of NO$_2$ TropVCD between CTRL and TM5 run (CTRL – TM5).

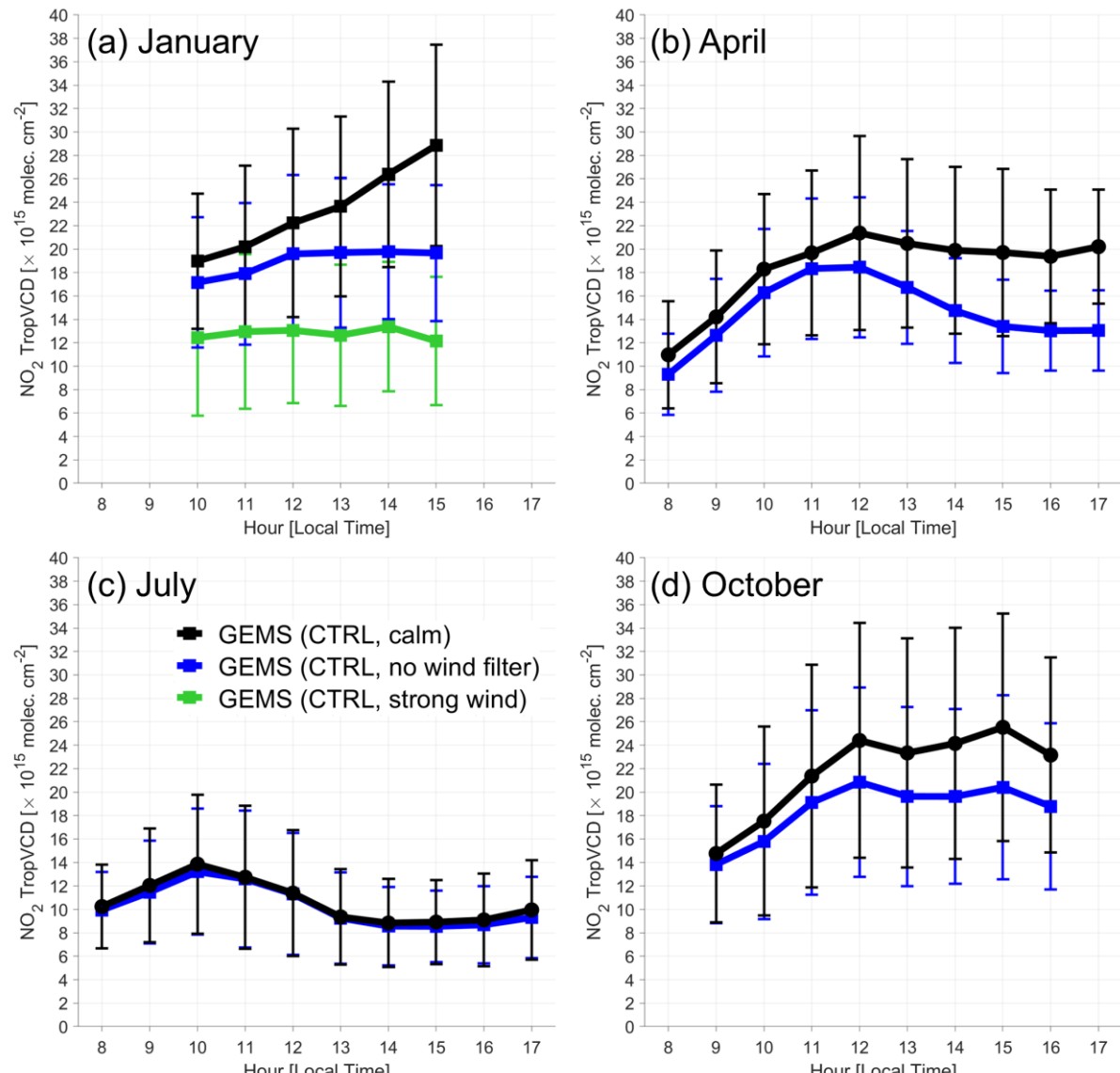

**Figure 12.** Diurnal patterns of retrieved NO$_2$ TropVCDs from the CTRL run in (a)

January, (b) April, (c) July, and (d) October 2021 over the SMA region. Black lines

indicate the NO$_2$ TropVCD values with wind-filtered data; only the scenes with wind

speed lower than 3m/s are utilized. Blue lines are the averaged values without any wind

filters. The green line is for case of strong-wind run with the NO$_2$ TropVCD being

selected and averaged for wind speeds faster than 5m/s in January.