# Peer review of "Diurnal variations of NO2 tropospheric vertical column density over the Seoul"

_Atmospheric Measurement Techniques, 2024_

## Referee Comment (RC3)

**Review:**

**Seunghwan Seo et al.,** Diurnal variations of NO$_2$ tropospheric vertical column density over the Seoul Metropolitan Area from the Geostationary Environment Monitoring Spectrometer (GEMS): seasonal differences and impacts of varying *a priori* NO$_2$ profile data

**Summary:**

In this paper, the authors show the diurnal variation in tropospheric NO$_2$ over Seoul, derived from an analysis of GEMS data. They also compare the results to output from two chemical transport models and conduct a sensitivity study of the GEMS results to *a priori* inputs from the models. The scope of the study is fairly limited, but the methods appear generally sound and the conclusions supported. I do have some questions about the retrieval algorithm used, for example in regard to cloud treatment.

The manuscript is well-written and organized with appropriate references. Needed improvements mostly involve clarification and a more detailed description of the approach. However, these concerns are generally minor, and if addressed, I can recommend the paper for publication.

**General comments:**

(1) For an overview of the IUP algorithm, only an unpublished study (Richter et al) is cited. Published references are given for some of the details including features fit to the spectra (Ring, NO$_2$ with temperature correction and other trace gases), a radiative transfer model, LER reflection values, etc. Most of these are fairly standard in DOAS retrievals. The stratosphere-estimation algorithm of Beirle et al. is also cited, but applying this to a GEO satellite is non-trivial. The authors should provide more details of the GEMS NO$_2$ retrieval used here, including an explanation of the cloud correction/screening.

(2) What is the temporal data domain for the study? Dates are given in figures 5, 6, 7, but elsewhere are GEMS retrievals from all days of each month combined into monthly/hourly means? Was the same done for the model output? What cloud screening was used, if any? For completeness, please also give the year.

(3) I'm confused by the f2 line (yellow) in figure 6. If a fixed NO$_2$ profile is used, shouldn't the TropVCD in the model be constant by definition? Why are there differences in the afternoon? I'd suggest eliminating the f2 line, unless I'm misunderstanding what it means, in which case some explanation should be added.

**Minor comments and suggested corrections:**

(1) Page 2, Lines 10, 17: Please state what the ranges represent.

(2) Page 4, Line 3: Might be clearer stated as "…$NO_2$ TropVCD between the WRF-Chem- and TM5-based GEMS datasets…"

(3) Page 5, Line 18: "…chemistry scheme follows the Regional Atmospheric Chemistry Mechanism …"

(4) Page 5, Line 24: "…were combined with…"

(5) Page 7, Line 7: "…may affect $NO_2$ TropVCD values for each month."

(6) Page 7, Line 15: "VCDs from the two GEMS products were similar throughout…"

(7) Page 7, Line 20: "For all times…"

(8) Page 8, Line 2: "The differences in GEMS…"

(9) Page 8, Line 18: "Figure 5 compares the diurnal changes in GEMS $NO_2$ TropVCD…"

(10) Page 8, Line 21: "…the two GEMS data products…"

(11) Page 9, Lines 4-5: Giving a range here is confusing. I suggest "Therefore, $NO_2$ TropVCDs calculated using WRF-Chem f2 show values up to 16.5% lower before 13:45 KST and up to 4.9% higher…"

(12) Page 9, Lines 6-8: "Notably, despite the diverse diurnal variations in *a priori* data from TM5 and WRF-Chem v3, the retrieved columns based on these data exhibited similar diurnal patterns…"

(13) Figure 2: This figure is hard to read because the maps are small. Is it possible to expand them? Also, the label on the color scale is too small to see clearly. It would help to add "WRF-Chem v3 minus TM5" in the figure caption(s).

(14) Figure 6: Consider omitting the yellow line (f2). See "General comments".

---

## Author Response (AR1)

**Response to Reviewer 1's Comments**

We deeply appreciate the reviewer for comments that develop our manuscript substantially. The reviewer's comments are written in blue, and our responses are in black. Following the reviewers' comments, we revised our manuscript – major changes are summarized below.

(1) We replaced the emission inventory used for WRF-Chem simulation from EDGAR-HTAP version 3 to AQNEA version 2, with additional 20% of $NO_x$ reduction over South Korea to reflect decreasing trends of $NO_2$ between 2019 and 2021.

(2) We also updated the GEMS IUP-UB products from version 0.9 to version 1.0, which includes considerations for the effects of clouds.

(3) To clarify the content, we reorganized the experiment into five cases: TM5, CTRL, CONST, FINE, and MIXED. Further details can be found in Section 2.2.

One-on-one responses and information on the revision are given below.

Major points for improvement suggestions:
* * *
1) it is well known that it is not the a priori profile itself but its shape that is relevant for DOAS-type of retrievals (ie, see Palmer et al., 2011; Eskes and Boersma 2003 or discussions in Yang et al. 2023a), so it would be nice to compare in a quantitative way the different a priori profiles shapes, and not only their VCD and the curtain plots as done now in Figures 3 and 4. I suggest to use a quantitative estimate of the height that contains the larger part of the profile (see details below).

➔ In the revised manuscript, we utilized the updated model results and updated GEMS tropospheric $NO_2$ retrievals. Now $NO_2$ concentrations and vertical columns in WRF-Chem and TM5 are closer (compared to the previous version of manuscript) and profile shapes and $NO_2$ concentrations at low level are also similar. The air mass factors averaged for SMA are similar for January, April, and October. In July, the air mass factor (AMF) in the CTRL case is lower than that in the TM5 case. Please refer to Figure R1 below. We agree to the point that the profile shape is a main factor. However, Kim et al. (2018) reported that for the chemical species that are abundant mainly in the boundary layer, in practice, the air mass factor was inversely proportional to the near-surface concentrations (Figure R2). Kim et al. (2018) analyzed AMF of HCHO. But this principle can apply to $NO_2$ as well.

[Figure]

Figure R1. Diurnal patterns of the air mass factor during weekdays in (a) January, (b) April, (c) July, and (d) October 2021 over the SMA region. Gray lines indicate the TM5 case, while black lines mean the CTRL case. The pixels with wind speed faster than 3m/s are excluded.

[Figure]

Figure R2. The relationship between the AMF and model HCHO volume mixing ratio is demonstrated. Different colors denote different times. The HCHO mixing ratio at ~200 m a.g.l. is plotted (adapted from Kim et al., 2018).

Figure R3 and R4 show the profiles of $NO_2$ mixing ratio and shape factor from the TM5 and CTRL model results, respectively. We found the largest differences in the mixing ratios in July can be related to the largest differences in the air mass factors. Meanwhile, there are no substantial changes in the shape factors among seasons. Combination of this shape factor and scattering weight (or box AMF) ultimately determines AMF. We do not have available scattering weight profile (or box AMF) for further analysis.

[Figure]

**Figure R3.** Vertical profiles of *a priori* NO$_2$ mixing ratios at 08, 10, 12, 14, and 16 LT from the TM5 (gray) and CTRL (black) cases in January, April, July, and October 2021 over the SMA region.

[Figure]

**Figure R4.** Vertical profiles of shape factors at 08, 10, 12, 14, and 16 LT from the TM5 (gray) and CTRL (black) cases in January, April, July, and October 2021 over the SMA region.

2) the test with the emission changes is one of the key points of this paper, but several parts are not clear to me: why choose emissions from Los Angeles while emissions for Korea exist from KORUS-AQ campaigns or their modified version (as used in Yang et al. 2023a and 2023b and in Edwards et al 2024)? It should be explained. In addition, in particular the diurnal factor mentioned in P5, line 27: it would be nice to see what is this diurnal evolution (and not only refer to a paper), to link it with the VCD evolution, and also comment its difference with respect to the above-mentioned emissions estimates for Korea. Moreover, I am not an expert on emissions, but I am wondering if 20% change is really a "big" change: in several publications larger changes are seen over the day (see my detailed comment below for P5, line 27).

I would explain more here the reasons of these choices for the emission changes tests and help the reader in understanding why these changes do not "appear" in the corresponding models and GEMS VCD. Is this because the test is too restrictive/not realistic or is this related to the WRF-Chem model (version) vertical mixing that is not strong enough to allow surface differences to also have an impact on the tropospheric column?

I am surprised that 20% changes in surface (emissions) would not affect the Planetary Boundary Layer so much, to only change the $NO_2$ profile shape so little that this is almost not seen between v2 and v3 test (or am I missing something?).

➔ We are sorry for the confusion caused. We did not used the emission "values" from Los Angeles, but only applied "normalized diurnal variabilities" of Los Angeles Basin to reflect hourly changes of $NO_x$ emissions over the urban regions. As we confirmed in the previous study by Kim et al. (2024), applying the diurnal variation pattern from Los Angeles Basin to the SMA region demonstrated good simulation performance. Thus, we applied the same approach in this study. The diurnal profiles from the KORUS-AQ V5 emission inventory are similar to the one in Kim et al (2024). To avoid further confusion, we revised the text in Section 2.2 and added a figure to compare the diurnal variabilities for CTRL and TM5 cases (**Figure R5**).

[Figure]

**Figure R5.** Diurnal variabilities of normalized NOx emissions for CTRL (black) and TM5 (gray) cases over the SMA region.

In the revised manuscript, we utilized the updated model and GEMS retrievals. The model results are quite different from the previous version due to changes in emission inventory and emission pre-processor. 20% reductions just reflect potential emission changes between 2018 and 2021 (previous version of the manuscript) or between 2019 and 2021 (revised version).

3) the AMF shown in Fig. 7 and 8 are all smaller than 1, with some diurnal and seasonal changes and different magnitude for 2 of the models used as a priori profile. When looking at other papers in the special issue, and esp. Yang et al. 2023a Figure 6 and Table 1, I am surprised how different are the AMF values. For Korus AQ conditions, May-June 2016, also over SMA region, they calculate AMF values always larger than 1. Please comment on the differences and ideally also add on Fig 8 the geometrical AMF (the Yang et al calculation is between 2.5 and 3), so that we could compare comparable quantities (only the part related to geometry and not the choice of the model used as a priori). The profile shape should be very different, with surface

(or PBL) contribution probably much more peaked in your case compared to what they measure/use from KORUS-AQ ?

➔ We thoroughly investigated several possible reasons for the differences between our AMF values and those reported in other studies, such as Yang et al. (2023). As shown in **Figure R6**, differences in the *a priori* data did not lead to significant changes in AMF values, except for July. On the other hand, the key differences between the results by Yang et al. (2023) and our study can be summarized as follows: (1) the surface albedo data, (2) the definition of the Seoul Metropolitan Area (SMA), and (3) cloud conditions. We utilized the surface albedo based on the TROPOMI Lambertian equivalent reflectivity (LER) climatology, while Yang et al. chose the OMI Level 3 LER climatology. Yang et al.'s study used a surface albedo value of 0.075, whereas our study applied values around 0.052 to 0.056 for July. As discussed in Section 5.4 of Lange et al. (2024), differences of surface reflectivity can cause a big change of retrieved VCDs. The different definition of SMA also should be considered. In this study, we defined the SMA as 126.5°E to 127.3°E and 37.2°N to 37.8°N. In contrast, Yang et al. (2023) defined it as 126.6°E to 127.7°E and 37.0°N to 37.6°N, which covers a larger domain than ours. Also, Yang et al. (2023) focused on clear-sky conditions while cloud impacts are considered in this study. When we adjust the definition of SMA into the Yang et al.'s domain and focused on clear-sky conditions, the AMF values increased by up to 14.0% (not shown here).

Our model setting was tested against the KORUS-AQ aircraft observations in Kim et al (2024). The model vertical profiles agreed well with the aircraft observations. This kind of comparison is useful, but it has also limitations due that aircraft cannot measure the composition in the lower level of atmosphere.

[Figure]

**Figure R6.** Diurnal patterns of retrieved (solid) and *a priori* (dashed) NO₂ TropVCDs during weekdays in (a) January, (b) April, (c) July, and (d) October 2021 over the SMA region. Gray lines indicate the TM5 case, while black lines mean the CTRL case. The pixels with wind speed faster than 3m/s are excluded.

4) the WRF-Chem model has a higher resolution (28x28km2) than the TM5 model (1°x1°) model used in TROPOMI retrievals. It would be interesting to average the WRF-Chem model to the 1°x1° resolution, to quantify the dilution effect mentioned in p.7, line 23.

➔ Thanks for your suggestion. We added **Figure R7** that compares regridded GEMS NO₂

TropVCD values of CTRL and TM5 cases into the TM5 model grid. After regridding, CTRL case shows still higher values of NO$_2$ TropVCD than TM5 case (**Figure R7c**), but the amounts of differences over SMA are much smaller than those before regridding (**Figure 11** in the main article).

[Figure]

**Figure R7.** Spatial distributions of retrieved NO$_2$ TropVCDs at 09:45 LT, July 2021 with regridding into the TM5 model grid – (a) CTRL case, (b) TM5 case, and (c) the differences between CTRL and TM5 cases. Note that the color scales of figure are different to those of Figure 3 to 5 in the main article.

Minor points/specific comments:
* * *
- P2, line 14: "uniformly 20%-reduces": uniformly in space or in time? or both?

➔ We reduced NOx emissions by 20% uniformly in both space and time. However, we replaced the emission inventory from EDGAR-HTAP version 3 to AQNEA version 2, with reduction of 20% to consider yearly trends of NO$_2$ between 2019 and 2021.

- P2, line 16: say why 13h45 KST (it is TROPOMI overpass time!)

➜ As you mentioned, 13:45 KST is TROPOMI overpass time – it is familiar to compare the values with TROPOMI data. We have a plan to compare our results with TROPOMI, but do not include in this study since it is beyond the scope of this study.

- P4, line 24: version 0.9. It is good to have a clear info on the version, but as the paper is not yet available, it is not so easy to compare, eg with Lange et al 2024 paper in the same special issue, that also uses IUPB GEMS NO2 data. Is this the same version? If there is no cloud correction, what do you do? you filter based on GEMS cloud fraction? you keep all the points? Please specify.

➜ We update the version of GEMS IUP-UB products from version 0.9 to version 1.0 (same version with Lange et al.'s 2024 paper), which includes the cloud correction.

- P5, sect.  2.2: some details, as the number of layers are given, which, as is, do not bring much information. Give the number of layer wrt to Top and bottom of the atmosphere, or the number of layers within the troposphere? Are the tropopause definitions the same between the models?

➜ The number of vertical layers is from the surface to the top of the atmosphere. However, we applied same tropopause to compute AMF and retrieve TropVCD values for both two models.

- P5, line 27: "a diurnal factor" --> this is an important element in my opinion, which is not explained enough. Why is Los Angeles relevant for Korea? why shifting it by 1 hour? is the diurnal factor the same for every season? Is the diurnal evolution larger than the 20% changes used for test v3? It would be nice to have an illustration of the time evolution of this factor, in order to be able to compare it to other choices made by other papers in the special issue. Eg

diurnal emission pattern in Yang 2023b, or Edwards et al. 2023, using NIER/KU-CREATE inventory from KORUS-AQ filed campaign (fig2): there are up to 75% changes during the day. Jo et al 2023 uses MUSICA model with different resolutions, and there they use KORUS v5 inventory. There is a nice discussion about different emission estimates and incorporation of diurnal variation into existing monthly emission estimates. It would be nice to refer to these publications and comment the relevance of your choice and how much it is different from the others.

The 20% emission reduction test: 20% is maybe not a change "big enough"? in Yang et al. 2023b, the ratios of 2022/2015 in Korea are of 0.70 for DJF and 0.50 in JJA, so 30% and 50% reductions... please comment.

Table 1: add in the first line (v2) if it is by default with the diurnal factor.

See major comment n°2).

➔ As explained in the reply of the major comment #2, we applied "normalized" diurnal variabilities into the existing NOx emission inventory (**Figure R5**). We assumed that the diurnal variability in urban areas like Seoul and Los Angeles would be similar, though the absolute values are different. However, the diurnal variabilities are 1-hour shifted to reflect a difference of the time when human activity starts – we suppose that commuting time in Seoul is one-hour earlier than that in Los Angeles. There are no differences by seasonal changes – as shown in Figure 1 of Yang et al. (2024), the diurnal variabilities of NOx emissions over SMA are nearly identical in winter and summer. The diurnal variabilities we applied are similar to those from prior studies such as Jo et al. (2023), Edwards et al. (2024), and Yang et al. (2024).

**Figure R8** is the normalized yearly trend of surface-observed $NO_2$ concentration over South Korea (black solid) and SMA (red solid) from 2001 to 2022, obtained from the Airkorea website (KEC, 2023). Since the reference years of emission inventories we chose are 2018 (EDGAR-HTAP version 3) or 2019 (AQNEA version 2), we reduced NOx emissions by 20%.

[Figure]

**Figure R8.** The yearly trends of surface-observed $NO_2$ concentrations from 2001 to 2022 over South Korea (black solid) and SMA (red solid). $NO_2$ concentrations are normalized by the values in 2019.

- P7, line 21: add the pink box in Fig 2 also in a non-empty subplot (it would be easier to follow the discussion.)

Also relevant here: line 23 "The coarser horizontal resolution of the TM5 model would be one of the reasons why these differences occur": although I tend to agree with this statement, it is

too speculative. You have a model with higher resolution, so you could actually resample it to 1°x1° resolution and quantify how much of the seen difference is due to purely dilution effects related to the different resolution. See major point 4).

It would also be good to cite studies that tried to quantify NO2 dilution effects (with satellites or models).

➔ Instead of adding the pink box into the non-empty subplots, we zoomed-in the figures to capture the detailed features over the SMA and surrounding regions. We also include **Figure R7**. $NO_2$ dilution effects by different model resolution are described in Valin et al. (2011) – it has been reported that the $NO_2$ columns simulated by WRF-Chem decrease as the resolution becomes coarser.

- P8,  lines 2 to 9): this discussion lacks an important point. In DOAS-like satellite retrievals, as in GEMS, only the apriori profile SHAPE is important, not its magnitude. This should be mentioned here and discussed. I would recommend adding some plots of the profile shapes comparison between WRF-Chem (v2 and v3) and TM5, maybe through H75 diurnal variation comparisons (H75 is the height at which 75% of the profile is included, see eg Vlemmix et al. 2015 (see major comment n° 1)

➔ We added the vertical profile shapes of each *a priori* data into **Figure R3** and **R4** and also in the main article and discussed about them.

- P8, line 18: "diurnal profiles of GEMS" --> the term profile is misleading. I would suggest something like "diurnal VCD evolution", not to confuse the reader with the vertical profile.

➔ Thanks for your suggestion. We modified the term "diurnal profiles" to "diurnal variabilities" or "diurnal patterns" to prevent potential confusions of readers.

➔ In the previous manuscript, we studied the period of October 25 – 28, 2021 to investigate characteristics of GEMS data. However, we covered whole days of January, April, July and October in the revised manuscript.

- P8, line 23: model tropo VCD between v2 and v3 are nearly identical --> same model, different emissions: is the profile shape the same? (I suppose so, as no changes in GEMS VCD). This would mean that the model has too small vertical mixing? or that the 20% change emission is a too small change? please comment.

➔ In the pre-processing code using EDGAR-HTAP version 3, there was an error in mass-conserving interpolation routine, which caused large overestimation of $NO_x$ emissions and simulated $NO_2$ columns. Thus, 20% reduction might be too small to see the impacts. We also used a few days of the mode simulations for this analysis in the previous manuscript and only utilized the points where satellite data were available. In the revised manuscript, we adjusted the processing code and replaced the emission inventory from EDGAR-HTAP version 3 to the most up-to-date AQNEA version 2. In addition, we used the data covering one full month for analysis of satellite data and model results.

Also, I am wondering why there is a small reduction of the VCD at 14h for f3 (and the others) in Fig 6 (and thus also Fig 5), while it is constant for the rest of the day? How do you explain this?

➔ Small fluctuations that appeared at 14 LT are due to filtering – since we analyzed only four days, filtering over the SMA domain have relatively large impacts. However, we analyzed whole months in the revised manuscript.

- P9, line 2: "almost no temporal variations of AMF" --> this is the part related to viewing

angles, albedo, etcc, which do not change much during that period? adding the geometric AMF would help understanding better this.

➜ Although the profile shapes were constant over time in WRF-Chem f2 case, other elements such as solar zenith angle (SZA), relative azimuth angle (RRA), or scattering weight changed, which made small fluctuations of AMF. In the revised manuscript, we omit this comment.

- P9, line 6 to 8: "Notably, despite... " --> this is good: this is the purely SCD contribution, hopefully not all the VCD change is coming from the AMF! I would insist on this. What is the part of the VCD that is actually coming from the measurements (the SCD) and which one is from AMF? Is there a time (diurnal and seasonal) variability?

➜ It is quite complex to separate the impacts on VCDs from SCD and AMF. Averaged diurnal evolution patterns of retrieved TropVCD seem independent to those of simulated values (**Figure 2**). However, when we utilized the *a priori* data of specific time (like CONST case), the diurnal variabilities of retrieved VCDs changed significantly by up to 19% (**Figure 6a**). Meanwhile, spatial distributions of GEMS $NO_2$ columns (**Figure 11**) show differences stemming from *a priori* profiles.

- table 1: clarify if some of these tests have constant emission factor during the day or if all share the same diurnal factor. It is not clear to me. + add a plot of the emission evolution during the day.

➜ We reorganized the table that summarizes our experiment designs (**Table 1**). We also added a figure that displays the diurnal variabilities of NOx emissions (**Figure 1**).

- Fig1: can you comment on WRF_chem v2. Is it the same than v3 for all seasons? (in Fig 5 for a few days yes)

➔ The difference between WRF-Chem v2 and WRF-Chem v3 case was caused by the NOx emission inventory – NOx emissions of WRF-Chem v3 were 20% lower than those of WRF-Chem v2. However, retrieved VCDs were very close to each other since calculated AMFs were similar. In the revised manuscript, this part was omitted.

- Fig2: in April, for the first hours of measurements there seems to be a clear yellow band on the left of the map (figure is too small and is difficult to estimate the longitude) compared to bluish columns over Korea. Is this due to a resolution effect of the model? or the albedo? or a land/ocean effect? Please comment.

➔ We think that those distinguishments occurred due to the resolution differences – but the absence of cloud considerations also can affect those features. After updating GEMS IUP-UB products from version 0.9 to version 1.0, those features do not appear or lessen, which can be found in **Figure 11** of revised manuscript.

**References**

Bucsela, E. J., Krotkov, N. A., Celarier, E. A., Lamsal, L. N., Swartz, W. H., Bhartia, P. K., Boersma, K. F., Veefkind, J. P., Gleason, J. F., and Pickering, K. E.: A new stratospheric and tropospheric $NO_2$ retrieval algorithm for nadir-viewing satellite instruments: applications to OMI, *Atmos. Meas. Tech.*, 6, 2607-2626, https://doi.org/10.5194/amt-6-2607-2013, 2013.

Edwards, D. P., Martínez-Alonso, S., Jo, D. S., Ortega, I., Emmons, L. K., Orlando, J. J., Worden, H. M., Kim, J., Lee, H., Park, J., and Hong, H.: Quantifying the diurnal variation of atmospheric $NO_2$ from observations of the Geostationary Environment Monitoring Spectrometer (GEMS), *Atmos. Chem. Phys.*, 24, 8943-8961, https://doi.org/10.5194/acp-24-8943-2024, 2024.

Jo, D. S., Emmons, L. K., Callaghan, P., Tilmes, S., Woo, J.-H., Kim, Y., Kim, J., Granier, C., Soulié, A., Doumbia, T., Darras, S., Buchholz, R. R., Simpson, I. J., Blake, D. R., Wisthaler, A., Schroeder, J. R., Fried, A., and Kanaya, Y.: Comparison of Urban Air Quality Simulations During the KORUS-AQ Campaign With Regionally Refined Versus Global Uniform Grids in the Multi-Scale Infrastructure for Chemistry and Aerosols (MUSICA) Version 0, *Journal of Advances in Modeling Earth Systems*, 15, e2022MS003458, https://doi.org/10.1029/2022MS003458, 2023.

KEC: The Korea Environment Corporation $NO_2$ dataset in South Korea – Airkorea [dataset], https://www.airkorea.or.kr, 2023.

Kim, K.-M., Kim, S.-W., Seo, S., Blake, D. R., Cho, S., Crawford, J. H., Emmons, L. K., Fried, A., Herman, J. R., Hong, J., Jung, J., Pfister, G. G., Weinheimer, A. J., Woo, J.-H., and Zhang, Q.: Sensitivity of the WRF-Chem v4.4 simulations of ozone and formaldehyde and their precursors to multiple bottom-up emission inventories over East Asia during the KORUS-AQ 2016 field campaign, *Geosci. Model Dev.*, 17, 1931-1955, https://doi.org/10.5194/gmd-17-1931-2024, 2024.

Kim, S.-W., Natraj, V., Lee, S., Kwon, H.-A., Park, R., de Gouw, J., Frost, G., Kim, J., Stutz, J., Trainer, M., Tsai, C., and Warneke, C.: Impact of high-resolution a priori profiles on satellite-based formaldehyde retrievals, Atmos. Chem. Phys., 18, 7639–7655, https://doi.org/10.5194/acp-18-7639-2018, 2018.

Kim, S.-W., McDonald, B. C., Brown, S. S., Dube, B., Ferrare, R. A., Frost, G. J., Harley, R. A., Holloway, J. S., Lee, H.-J., McKeen, S. A., Neuman, J. A., Nowak, J. B., Oetjen, H., Ortega, I., Pollack, I. B., Roberts, J. M., Ryerson, T. B., Scarino, A. J., Senff, C. J., Thalman, R., Trainer, M., Volkamer, R., Wagner, N., Washenfelder, R. A., Waxman, E., and Young, C. J.: Modeling the weekly cycle of $NO_x$ and CO emissions and their impacts on $O_3$ in the Los Angeles-South Coast Air Basin during the CalNex 2010 field campaign, *J. Geophys. Res. Atmos.*, 121, 1340-1360, https://doi.org/10.1002/2015JD024292, 2016.

Lange, K., Richter, A., Bösch, T., Zilker, B., Latsch, M., Behrens, L. K., Okafor, C. M., Bösch, H., Burrows, J. P., Merlaud, A., Pinardi, G., Fayt, C., Friedrich, M. M., Dimitropoulou, E., Van Roozendael, M., Ziegler, S., Ripperger-Lukosiunaite, S., Kuhn, L., Lauster, B., Wagner, T., Hong, H., Kim, D., Chang, L.-S., Bae, K., Song, C.-K., and Lee, H.: Validation of GEMS tropospheric $NO_2$ columns and their diurnal variation with ground-based DOAS measurements, *egusphere [preprint]*, https://doi.org/10.5194/egusphere-2024-617, 2024.

Valin, L. C., Russell, A. R., Hudman, R. C., and Cohen, R. C.: Effects of model resolution on the interpretation of satellite $NO_2$ observations, *Atmos. Chem. Phys.*, 11, 11647-11655, https://doi.org/10.5194/11-11647-2011, 2011.

Vlemmix, T., Hendrick, F., Pinardi, G., De Smedt, I., Fayt, C., Hermans, C., Piters, A., Wang, P., Levelt, P., and Van Roozendael, M.: MAX-DOAS observations of aerosols, formaldehyde and nitrogen dioxide in the Beijing area: comparison of two profile retrieval approaches, *Atmos. Meas. Tech.*, 8, 941-963, https://doi.org/10.5194/amt-8-941-2015, 2015.

Yang, L. H., Jacob, D. J., Colombi, N. K., Zhai, S., Bates, K. H., Shah, V., Beaudry, E., Yantosca, R. M., Lin, H., Brewer, J. F., Chong, H., Travis, K. R., Crawford, J. H., Lamsal, L. N., Koo,

J.-H., and Kim, J.: Tropospheric $NO_2$ vertical profiles over South Korea and their relation to oxidant chemistry: implications for geostationary satellite retrievals and the observations of $NO_2$ diurnal variation from space, *Atmos. Chem. Phys.*, 23, 2465-2481, https://doi.org/10.5194/acp-23-2465-2023, 2023.

Yang, L. H., Jacob, D. J., Dang, R., Oak, Y. J., Lin, H., Kim, J., Zhai, S., Colombi, N. K., Pendergrass, D. C., Beaudry, E., Shah, V., Feng, X., Yantosca, R. M., Chong, H., Park, J., Lee, H., Lee, W.-J., Kim, S., Kim, E., Travis, K. R., Crawford, J. H., and Liao, H.: Interpreting Geostationary Environment Monitoring Spectrometer (GEMS) geostationary satellite observations of the diurnal variation in nitrogen dioxide ($NO_2$) over East Asia, *Atmos. Chem. Phys.*, 24, 7027-7039, https://doi.or/10.5194/acp-24-7027-2024, 2024.

**Response to Reviewer 2's Comments**

We deeply appreciate the reviewer for comments that develop our manuscript substantially. The reviewer's comments are written in blue, and our responses are in black. Following the reviewers' comments, we revised our manuscript – major changes are summarized below.

(1) We replaced the emission inventory used for WRF-Chem simulation from EDGAR-HTAP version 3 to AQNEA version 2, with additional 20% reduction of NOx emissions over South Korea to reflect decreasing trends of $NO_2$ between 2019 and 2021.

(2) We also updated the GEMS IUP-UB products from version 0.9 to version 1.0, which includes considerations for the effects of clouds.

(3) To clarify the content, we reorganized the experiment into five cases: TM5, CTRL, CONST, FINE, and MIXED. Further details can be found in Section 2.2.

One-on-one responses and information on the revision are given below.

1. Clouds have a strong impact on the visibility of tropospheric NO2 from space. This is shortly mentioned in the manuscript (Page 3, Line 20), but nevertheless ignored in the analysis (Page 4, Line 25).

Clouds potentially affect both key topics of the paper:

- Cloud parameters vary over the day as well. This affects visibility (i.e., AMFs), and can lead to virtual diurnal patterns in the NO2 column if not accounted for properly.

- Cloud effects are coupled to the vertical profile, as they shield the column below, but increase the visibility above.

Thus, depending on cloud height, I would expect that for partly clouded pixels the AMF variations for different a-priori profiles can be much larger than those shown in Fig. 8, and the conclusion that "different a-priori profiles ... have only minimal impact" is potentially misleading.

Thus, this study needs (a) to explicitly consider, or (b) at least throroughly (and quantitatively) discuss cloud effects.

➔ We agree to the reviewer's comment regarding the importance of cloud effects. To consider the impacts of clouds, we updated GEMS IUP-UB products from version 0.9 to version 1.0 (Lange et al., 2024). In the updated version, the adjusted cloud fractions and pressure from the GEMS L2 cloud product were adopted to apply the AMF cloud correction. Also, cloudy scenes were excluded by using pixels with qa_value higher than 0.75.

2. The WRF simulations do not match the observed columns, but are far higher.

Before using WRF as kind of reference for investigating the impact of vertical profiles, this issue should be fixed (which is probably related to far too high emissions, definitely more than the 20% investigated in the manuscript).

Due to these issues, the manuscript needs extensive major revisions, which will also affect the drawn conclusions. Thus I do not provide detailed comments on the text at this stage, but an additional iteration for review will be required after the manuscript has been revised accordingly.

➔ We found an error in the code that pre-processes the emission inventory. We improved the

processing code and also replaced the emission inventory from EDGAR-HTAP version 3 to recently updated AQNEA version 2, with additional 20% reduction of NOx emissions to consider the changing trends between 2019 and 2021. **Figure R1** shows diurnal patterns of retrieved (solid) and simulated (dashed) $NO_2$ TropVCDs from CTRL and TM5 cases. VCD values simulated from WRF-Chem become significantly closer to the retrieved values compared to the previous version of the manuscript. In the previous version, WRF-Chem shows up to 12 times higher values of simulated $NO_2$ TropVCDs than TM5 in July. However, in the revised version, WRF-Chem $NO_2$ TropVCDs are up to 2.75 times higher than TM5 $NO_2$ TropVCDs.

[Figure]

**Figure R1.** Diurnal patterns of retrieved (solid) and *a priori* (dashed) $NO_2$ TropVCDs during weekdays in (a) January, (b) April, (c) July, and (d) October 2021 over the SMA region. Gray lines indicate the TM5 case, while black lines mean the CTRL case. The pixels with wind speed faster than 3m/s are excluded.

Additional comments:

- Page 2 Line 11: "decrease over time" is potentially misleading, as it might be understood as a temporal trend, while I think it is meant to describe the diurnal patterns. This should be clarified (e.g. by adding "diurnal").

➔ We modified the term you mentioned.

- Page 3 Line 20: terrain height is not varying.

➔ We modified the term you mentioned from "varying ancillary data" to "several ancillary data".

- Page 5 Line 4: what is different in the "variant" from STREAM?

➔ The changes from STREAM to STREAM-B are:

   (1) the widths of the kernels as well as the thresholds used have been adapted to GEMS data,

   (2) we do not use scenes with high clouds as this is not reliable with the current cloud height product,

   (3) the masking of regions with tropospheric pollution is based on the TM5 tropospheric columns for the same day and time, not a climatology,

   (4) there is a correction for free tropospheric $NO_2$ applied based on the tropospheric columns of the TM5 model over "clean" regions.

   Further details will be provided in Richter et al. (2024, *in preparation*).

- Figures: The diurnal cycle of NO2 is prominent topic of this study, but it is actually not shown;

Fig. 2 displays differences of columns for different models, but not the column itself.

I would recommend to add a figure showing the diurnal cycle of VCD maps. For sake of readability, I would recommend to show only every second hour. A similar figure of the diurnal patterns of cloud fraction and cloud height might be added as well.

➔ Thanks for your suggestion. We added spatial distributions of VCDs for every two hours in **Figure R2** to **R4** here and **Figures** in the main article.

[Figure]

**Figure R2.** Spatial distributions of retrieved NO₂ TropVCDs in January, April, July, and October 2021 from the TM5 case. The pixels with wind speed faster than 3m/s are excluded.

[Figure]

**Figure R3.** Same as Figure R2, except from the CTRL case.

[Figure]

**Figure R4.** Same as Figure R2, except for the differences between CTRL and TM5 case.

- I don't understand Fig. 6: f2 is a scenario with fixed WRF profile over the day. How can this fixed profile show diurnal variations in the modeled column?

➔ For the fair comparison between satellite and model values, we also excluded model values for pixels in which satellite observations are unavailable. As a result, diurnal variability may still be present even in the model values, especially for the short analysis period (October 25 – 28, 2021) in the previous manuscript.

**References**

Beirle, S., Hörmann, C., Jöckel, P., Liu, S., Penning De Vries, M., Pozzer, A., Sihler, H., Valks, P. and Wagner, T.: The STRatospheric Estimation Algorithm from Mainz (STREAM): Estimating stratospheric NO2 from nadir-viewing satellites by weighted convolution, *Atmos. Meas. Tech.*, 9(7), 2753–2779, https://doi.org/10.5194/amt-9-2753-2016, 2016.

Lange, K., Richter, A., Bösch, T., Zilker, B., Latsch, M., Behrens, L. K., Okafor, C. M., Bösch, H., Burrows, J. P., Merlaud, A., Pinardi, G., Fayt, C., Friedrich, M. M., Dimitropoulou, E., Van Roozendael, M., Ziegler, S., Ripperger-Lukosiunaite, S., Kuhn, L., Lauster, B., Wagner, T., Hong, H., Kim, D., Chang, L.-S., Bae, K., Song, C.-K., and Lee, H.: Validation of GEMS tropospheric $NO_2$ columns and their diurnal variation with ground-based DOAS measurements, *egusphere [preprint]*, https://doi.org/10.5194/egusphere-2024-617, 2024.

**Response to Reviewer 3's Comments**

We deeply appreciate the reviewer for comments that develop our manuscript substantially. The reviewer's comments are written in blue, and our responses are in black. Following the reviewers' comments, we revised our manuscript – major changes are summarized below.

(1) We replaced the emission inventory used for WRF-Chem simulation from EDGAR-HTAP version 3 to AQNEA version 2, with additional 20% reduction of NOx emissions over South Korea to reflect decreasing trends of $NO_2$ between 2019 and 2021.

(2) We also updated the GEMS IUP-UB products from version 0.9 to version 1.0, which includes considerations for the effects of clouds.

(3) To clarify the content, we reorganized the experiment into five cases: TM5, CTRL, CONST, FINE, and MIXED. Further details can be found in Section 2.2.

One-on-one responses and information on the revision are given below.

**General comments:**

(1) For an overview of the IUP algorithm, only an unpublished study (Richter et al) is cited. Published references are given for some of the details including features fit to the spectra (Ring, NO2 with temperature correction and other trace gases), a radiative transfer model, LER reflection values, etc. Most of these are fairly standard in DOAS retrievals. The stratosphere-estimation algorithm of Beirle et al. is also cited, but applying this to a GEO satellite is nontrivial. The authors should provide more details of the GEMS $NO_2$ retrieval used here, including an explanation of the cloud correction/screening.

➔ To consider the cloud correction, we updated the GEMS IUP-UB products from version 0.9 to version 1.0 (Lange et al., 2024). In the updated version, adjusted cloud fractions and pressure from the GEMS L2 cloud product were adopted to apply the AMF cloud correction. Also, cloudy scenes were excluded by using pixels with qa_value higher than 0.75.

(2) What is the temporal data domain for the study? Dates are given in figures 5, 6, 7, but elsewhere are GEMS retrievals from all days of each month combined into monthly/hourly means? Was the same done for the model output? What cloud screening was used, if any? For completeness, please also give the year.

➔ In the previous version of the manuscript, we investigated all days of January, April, July, and October 2021 for the comparison of WRF-Chem v3 and TM5, while only four days (October 25 – 28, 2021) were analyzed for comparing WRF-Chem v2, f2, and v3. Also, there were no cloud considerations in the previous version.

In the revised manuscript, the data from all days of January, April, July, and October 2021 were analyzed to compare the CTRL and TM5 cases, and the data from all days of July 2021 were investigated to compare CTRL, CONST, FINE, and MIXED cases. By updating the GEMS IUP-UB products from version 0.9 to version 1.0, cloud corrections are included by using cloud pressure from the GEMS L2 cloud product and recomputed cloud fractions by SCIATRAN (Lange et al., 2024). We also excluded cloudy scenes with cloud radiance fraction higher than 0.5.

(3) I'm confused by the f2 line (yellow) in figure 6. If a fixed NO2 profile is used, shouldn't the TropVCD in the model be constant by definition? Why are there differences in the afternoon? I'd suggest eliminating the f2 line, unless I'm misunderstanding what it means, in which case

some explanation should be added.

➔ The values of TropVCD in the model surely did not change by time. Since we excluded the model pixels where observations do not exist, the domain-averaged model values can be fluctuated. Especially, **Figure 6** in the previous version of the manuscript shows the mean values during only four days (October 25 – 28, 2021); masking of each pixel can affect the mean values. In the revised paper, the fluctuation of model TropVCD from CONST case is also captured by the same reason. We added related explanation into the caption of Figure 6 in the revised manuscript.

**Minor comments and suggested corrections:**

We appreciate your suggested corrections. Since this paper has been majorly revised, many sentences you suggested for corrections have been deleted or rewritten. However, we have incorporated your suggested edits as much as possible during this process.

(1) Page 2, Lines 10, 17: Please state what the ranges represent.

➔ The ranges represent the maximum values in July from TM5 case to WRF-Chem v3 case.

(2) Page 4, Line 3: Might be clearer stated as "…NO2 TropVCD between the WRF-Chem- and TM5-based GEMS datasets…"

➔ Changed.

(3) Page 5, Line 18: "…chemistry scheme follows the Regional Atmospheric Chemistry Mechanism …"

➔ Changed (Page 5, Line 26).

(4) Page 5, Line 24: "…were combined with…"

➔ Changed (Page 6, Line 4).

(5) Page 7, Line 7: "…may affect NO2 TropVCD values for each month."

➔ The sentence is deleted in the revised manuscript.

(6) Page 7, Line 15: "VCDs from the two GEMS products were similar throughout…"

➔ The sentence is deleted in the revised manuscript.

(7) Page 7, Line 20: "For all times…"

➔ The sentence is deleted in the revised manuscript.

(8) Page 8, Line 2: "The differences in GEMS…"

➔ The sentence is deleted in the revised manuscript.

(9) Page 8, Line 18: "Figure 5 compares the diurnal changes in GEMS NO2 TropVCD…"

➔ The sentence is deleted in the revised manuscript.

(10) Page 8, Line 21: "…the two GEMS data products…"

➔ The sentence is deleted in the revised manuscript.

(11) Page 9, Lines 4-5: Giving a range here is confusing. I suggest "Therefore, NO2 TropVCDs calculated using WRF-Chem f2 show values up to 16.5% lower before 13:45 KST and up to 4.9% higher…"

➔ The sentence is deleted in the revised manuscript, but I refer to your comment to prevent any further confusion during the revision.

(12) Page 9, Lines 6-8: "Notably, despite the diverse diurnal variations in a priori data from TM5 and WRF-Chem v3, the retrieved columns based on these data exhibited similar diurnal patterns…"

➔ The sentence is deleted in the revised manuscript.

(13) Figure 2: This figure is hard to read because the maps are small. Is it possible to expand them? Also, the label on the color scale is too small to see clearly. It would help to add "WRFChem v3 minus TM5" in the figure caption(s).

➔ The expanded and zoomed-in maps are displayed in Figure 9 to Figure 11 in the revised manuscript. Also, we added "(CTRL – TM5)" in the captions of Figure 3 and Figure 11 in the revised manuscript.

(14) Figure 6: Consider omitting the yellow line (f2). See "General comments".

➔ As mentioned in the General comments #3, the diurnal changes of 'WRF-Chem f2' case in the previous version or 'CONST' case in the revised version occur during calculating domain-averaged values – the location and number of pixels excluded during the collocation with satellite data vary over time during the day.

**References**

Lange, K., Richter, A., Bösch, T., Zilker, B., Latsch, M., Behrens, L. K., Okafor, C. M., Bösch, H., Burrows, J. P., Merlaud, A., Pinardi, G., Fayt, C., Friedrich, M. M., Dimitropoulou, E., Van Roozendael, M., Ziegler, S., Ripperger-Lukosiunaite, S., Kuhn, L., Lauster, B., Wagner, T., Hong, H., Kim, D., Chang, L.-S., Bae, K., Song, C.-K., and Lee, H.: Validation of GEMS tropospheric $NO_2$ columns and their diurnal variation with ground-based DOAS measurements, *egusphere [preprint]*, https://doi.org/10.5194/egusphere-2024-617, 2024.